# Multi-Stage Predict+Optimize for (Mixed Integer) Linear Programs

**Xinyi Hu[1], Jasper C.H. Lee[2], Jimmy H.M. Lee[1], Peter J. Stuckey[3]**
[1]Department of Computer Science and Engineering,
The Chinese University of Hong Kong, Shatin, N.T., Hong Kong
[2]Department of Computer Science, University of California, Davis, USA
[3]Department of Data Science and AI, Monash University, Clayton, Australia
{xyhu,jlee}@cse.cuhk.edu.hk, jasperlee@ucdavis.edu, peter.stuckey@monash.edu

## Abstract

The recently-proposed framework of Predict+Optimize tackles optimization problems with parameters that are unknown at solving time, in a supervised learning setting. Prior frameworks consider only the scenario where *all* unknown parameters are (eventually) revealed at the same time. In this work, we propose *Multi-Stage Predict+Optimize*, a novel extension catering to applications where unknown parameters are instead revealed in sequential stages, with optimization decisions made in between. We further develop three training algorithms for neural networks (NNs) for our framework as proof of concept, all of which can handle mixed integer linear programs. The first baseline algorithm is a natural extension of prior work, training a single NN which makes a single prediction of unknown parameters. The second and third algorithms instead leverage the possibility of updating parameter predictions between stages, and trains one NN *per stage*. To handle the interdependency between the NNs, we adopt a sequential and parallelized versions of coordinate descent for training. Experimentation on three benchmarks demonstrates the superior learning performance of our methods over classical approaches.

## 1 Introduction

*Constrained optimization* problems can frequently model applications in everyday life, yet, the parameters of the problem are unknown at the time of solving. Consider, for example, a real-world application where hospital administrators need to schedule shifts for nurses, so as to minimize the total costs for hiring nurses while meeting the patient load. Here, the shifts need to be decided before the (real-time) patient demand is known, which requires *predicting* the demand when scheduling.

In the present work, we focus on the supervised learning setting, where unknown parameters can be predicted using relevant features, and historical (features, parameters) pairs serve as training data for the prediction model. The goal is to estimate the unknown parameters based on the related features, such that when we solve the optimization problem using the estimated parameters, the resulting solution is good even under the later-revealed true parameters.

Classic approaches, for example by learning predictors using (regularized) $\ell_2$ loss, do not necessarily work well — low prediction error in parameter space does not guarantee good performance of the estimated solution according to the optimization objective. The influential framework of Predict+Optimize proposed by Elmachtoub and Grigas [9] instead uses the more effective *regret loss*, which incorporates information about the optimization problem. However, the framework limits the unknown parameters to appear only in the objective and not the constraints — if uncertainty in constraints is mis-predicted, the resulting estimated solution might not even be feasible under the true parameters. Recent works by Hu et al. [15, 16] thus propose a *Two-Stage* adaptation of

Predict+Optimize, explicitly modelling 1) a prediction stage and 2) a recourse stage which corrects infeasible solutions into feasible ones. The new two-stage framework is therefore applicable even when the optimization constraints contain uncertainty.

However, the two-stage framework essentially assumes that all the unknown parameters are revealed simultaneously, excluding applications where such information is gradually released and new decisions need to be made across many stages (e.g. in a daily/weekly manner). Crucially, in these applications, predictions can also be updated between stages, in light of the new information and past committed actions. Consider again the example of scheduling shifts for nurses. A typical facility might have an appointment system, with reservations closing the day before each working day. As opposed to a two-stage modelling, a more practical approach would treat each day in a work week as its own stage where new information (the precise appointments the next day) is learned, inducing both new optimization decisions and updated predictions.

The concrete contributions of this paper are three-fold.

**Framework**    We propose and formalize the new framework of *Multi-Stage Predict+Optimize* (Section 3), where unknown parameters are revealed across multiple stages, inducing new optimization decisions and updated parameter predictions.

**Training algorithms**    The flexibility to update (future) parameter predictions in each stage introduces intricate challenges to the training process, which should train a prediction model *per stage*. The challenges are both in predictive power and in computation time. The performance of predictors across stages are *intertwined* and *interdependent*: the "goodness" of a prediction depends on actions in other stages, which in turn depends on predictions of those other stages. Such dependency can also cause serialization issues that could drastically lengthen training time.

In Section 4, we propose three neural network training algorithms for our framework, assuming that the optimization problems can be formulated as mixed integer linear programs (MILPs): 1) a baseline algorithm that directly generalizes the two-stage algorithm of Hu et al. [16], training only a single neural network predictor, 2) a sequential coordinate descent training algorithm which trains a neural network model *per stage*, and each stage is considered a "coordinate", and 3) a parallel version of coordinate descent. These algorithms trade off between training time and predictive performance.

**Empirical evaluation**    We apply these methods to three benchmarks (Section 5) to empirically demonstrate their superior learning performance over classical training methods, as well as the computation/prediction tradeoff between the proposed methods.

We note that there are other lines of work tackling similar settings, where unknown parameters in optimizations are also revealed in a sequential fashion. Perhaps the most well-known is multi-stage stochastic optimization [14, 30]. The main difference between our work and multi-stage stochastic optimization is *supervised* vs *unsupervised* learning. Our framework (and Predict+Optimize in general) has features that help making parameter predictions, whereas (non-contextual) stochastic optimization does not and requires different techniques to tackle. See Appendix A.3 for a detailed discussion on the connection and comparison between our framework and stochastic optimization.

**Related Work**    We include a brief literature review here in the main body. See Appendix A for a detailed exposition.

Elmachtoub and Grigas proposed the influential framework of Predict+Optimize [9], with lots of followup work in the community on improving computational efficiency [23, 24], predictive accuracy [7, 18, 22], on types of applicable optimization problems [12, 17, 36], and applying to specific real-world scenarios [6, 34, 35]. More recently, Hu et al. [15, 16] proposed adaptations of the framework to handle uncertainty in optimization constraints, including the Two-Stage framework which our work is most related to.

Predict+Optimize also sits in a broader line of work on *decision-focused learning*, including works that learn prediction models for unknown parameters but with different goals/loss measures [25, 28], as well as other works that invent techniques for differentiating through optimization problems [1, 2, 36].

Outside of decision-focused learning, our work is also somewhat related to (multi-stage) stochastic programming [14, 30]. The main difference, again, is supervised vs unsupervised learning. See Appendix A.3 for a more detailed comparison.

## 2   Background — Two-Stage Predict+Optimize

In this section, we describe Two-Stage Predict+Optimize [16], which is prerequisite to understanding our contributions. Without loss of generality, the framework is stated for minimization problems.

A *parameterized optimization problem (Para-OP)* $P(\theta)$ is defined as finding:

$$x^*(\theta) = \arg\min_x obj(x, \theta) \quad \text{s.t. } C(x, \theta)$$

where $x$ is a vector of decision variables, $\theta$ is a vector of parameters, $obj$ is a function mapping decisions $x$ and parameters $\theta$ to a real objective value that is to be minimized, and $C$ is a set of constraints that must be satisfied over $x$ under parameters $\theta$. We call $x^*(\theta)$ an *optimal solution* under parameters $\theta$, and $obj(x^*(\theta), \theta))$ the *optimal value* under parameters $\theta$. When the parameters are all known, a Para-OP is just a classical optimization problem (OP).

In the Predict+Optimize setting (from the original framework [9], to the two-stage framework [16], and to our multi-stage framework later on), each instantiation of the true parameter vector $\theta$ has an associated *feature matrix* $A$. These features are relevant information that can help a model predict $\theta$.

In the following, we number the stages by 0 and 1 in Two-Stage Predict+Optimize [16] to make the framework look more similar to the multi-stage framework we propose later in Section 3.

**Stage 0**   The practitioner uses a prediction model, which takes in a feature matrix $A$, to compute a vector of estimated parameters $\hat{\theta}$. The Stage 0 solution $\hat{x}_0$ is then computed as

$$\hat{x}^{(0)} = \arg\min_x obj(x, \hat{\theta}) \quad \text{s.t. } C(x, \hat{\theta})$$

The Stage 0 solution $\hat{x}^{(0)}$ should be interpreted as a form of soft commitment that can be modified in Stage 1, subject to a penalty.

**Stage 1**   The true parameters $\theta$ are revealed, and the practitioner wishes to compute an updated Stage 1 solution $\hat{x}^{(1)}$ from $\hat{x}^{(0)}$, subject to a (problem-specific) penalty function $Pen(\hat{x}^{(0)} \to \hat{x}^{(1)}, \theta)$ which depends on both the softly-committed Stage 0 $\hat{x}^{(0)}$, the final Stage 1 solution $\hat{x}^{(1)}$ and the true parameters $\theta$. More specifically, the Stage 1 solution $\hat{x}^{(1)}$ is computed as

$$\hat{x}^{(1)} = \arg\min_x obj(x, \theta) + Pen(\hat{x}^{(0)} \to x, \theta) \quad \text{s.t. } C(x, \theta)$$

The Stage 1 solution $\hat{x}^{(1)}$ should be interpreted as a hard-committed final action, and note that it is guaranteed to be feasible under the true parameters $\theta$.

The prediction $\hat{\theta}$ is evaluated using the *post-hoc regret* [16], which is the sum of two terms: (a) the difference in objective between the *true optimal solution* $x^*(\theta)$ and the final Stage 1 solution $\hat{x}^{(1)}$ under the true parameters $\theta$, and (b) the penalty incurred by modifying $\hat{x}^{(0)}$ to $\hat{x}^{(1)}$. Formally, the post-hoc regret function $PReg(\hat{\theta}, \theta)$ (for minimization problems) is:

$$PReg(\hat{\theta}, \theta) = obj(\hat{x}^{(1)}, \theta) - obj(x^*(\theta), \theta) + Pen(\hat{x}^{(0)} \to \hat{x}^{(1)}, \theta)$$

The goal of a prediction model is to make predictions $\hat{\theta}$ so as to minimize the post-hoc regret. We emphasize again that the main difference between Predict+Optimize frameworks and stochastic programming frameworks is that in Predict+Optimize, a prediction model has access to features relevant to the true parameters in order to make a prediction. Stochastic programming, on the other hand, frequently operates solely at the level of the distribution over the true parameters.

# 3  Multi-Stage Predict+Optimize

In this section, we present our new framework of *Multi-Stage* Predict+Optimize, which models applications where unknown parameters are revealed across $T$ different stages.

Consider again the Para-OP

$$\mathbf{x}^*(\boldsymbol{\theta}) = \arg\min_{\mathbf{x}} obj(\mathbf{x}, \boldsymbol{\theta}) \quad \text{s.t. } C(\mathbf{x}, \boldsymbol{\theta})$$

We view the true parameter vector $\boldsymbol{\theta}$ as $(\theta_1, \ldots, \theta_T)$, where each $\theta_t$ is the sub-vector of parameters released at Stage $t$. Similarly, we also view the vector of decision variables $\mathbf{x}$ as $(x_0, \ldots, x_T)$, where each $x_t$ is the sub-vector of decision variables that are hard-committed in Stage $t$ (e.g. via a concrete real-world action taken at Stage $t$) and soft-committed in prior stages (e.g. a tentative nurse schedule).

At a high level, the parameters $\theta_t$ are revealed at Stage $t$, and a model makes a prediction $\hat{\boldsymbol{\theta}}^{(t)} = (\hat{\theta}_{t+1}^{(t)}, \ldots, \hat{\theta}_T^{(t)})$ of all the *unrevealed* parameters. Then, the practitioner solves the Stage $t$ optimization problem which we define later in the section. The decision variables $x_t$ are newly hard-committed, whereas the decision variables $x_{t+1}, \ldots, x_T$ are soft-committed with potential to be modified in future stages (at the cost of a penalty). This process is repeated until all stages are completed.

In the rest of the section and paper, we will use the standard notation of $\boldsymbol{\theta}[t_1 : t_2] = (\theta_{t_1}, \ldots, \theta_{t_2})$ to denote sub-vectors (treated as arrays), and use the notation $\oplus$ for vector concatenation.

## 3.1  Formal Framework Definition

Now we formally define Multi-Stage Predict+Optimize framework. In Appendix B, we also present the hospital scenario from the Introduction as a detailed example of applying this framework.

**Stage 0**   None of the true parameters have been revealed. $Model_0$ takes the feature matrix $A$ and predicts $\hat{\boldsymbol{\theta}}^{(0)} = (\hat{\theta}_1^{(0)}, \ldots, \hat{\theta}_T^{(0)})$. The practitioner then computes the Stage 0 solution $\hat{x}^{(0)}$ as

$$\hat{x}^{(0)} = \arg\min_{x} obj(x, \hat{\boldsymbol{\theta}}^{(0)}) \quad \text{s.t. } C(x, \hat{\boldsymbol{\theta}}^{(0)})$$

The decision variables $\hat{x}_0^{(0)}$ are hard commitments, whereas the rest of the decision vector $\hat{x}_1^{(0)}, \ldots, \hat{x}_T^{(0)}$ are soft commitments.

**Stage $t$ (for $1 \leq t \leq T$)**   The true parameters $\theta_1, \ldots, \theta_{t-1}$ were previously revealed, and $\theta_t$ is newly revealed. $Model_t$ makes a prediction $\hat{\boldsymbol{\theta}}^{(t)} = (\hat{\theta}_{t+1}^{(t)}, \ldots, \hat{\theta}_T^{(t)})$ using 1) the feature matrix $A$, 2) the previous stage solution $\hat{x}^{(t-1)}$ and 3) the revealed true parameters $\theta_1, \ldots, \theta_t$. For computational efficiency reasons, $Model_t$ may instead take any subset or derived functions of the above inputs. For example, $Model_t$ can choose whether or not to incorporate the revealed true parameters $\theta_1, \ldots, \theta_t$ as input. While these revealed parameters can serve as additional features, potentially guiding and correcting current predictions more effectively, they also increase training time (and inference time to a smaller extent). The trade-off between prediction improvement and additional training time depends on the optimization problem, model structure, and training data. Therefore, whether to utilize the revealed true parameters can be considered a hyperparameter that should be tuned for each application using the available training data. See Appendix H.1 for a more detailed discussion.

Afterwards, the practitioner computes the Stage $t$ solution $\hat{x}^{(t)}$ using the following Stage $t$ optimization problem, which crucially modifies the original Para-OP by: 1) introducing a penalty term $Pen_t(\hat{x}^{(t-1)} \to x, \boldsymbol{\theta}[1 : t])$ capturing the cost of changing the Stage $t-1$ solution $\hat{x}^{(t-1)}$ to the new Stage $t$ solution $x$, and 2) introducing the constraint that $x[1 : t-1] = \hat{x}^{(t-1)}[1 : t-1]$, namely, hard commitments from prior stages cannot be modified in the current Stage $t$. This constraint is a form of a non-anticipativity constraint in the stochastic programming literature [14].

$$\hat{x}^{(t)} = \quad \arg\min_{x} obj(x, \boldsymbol{\theta}[1 : t] \oplus \hat{\boldsymbol{\theta}}^{(t)}) + Pen_t(\hat{x}^{(t-1)} \to x, \boldsymbol{\theta}[1 : t])$$

$$\text{s.t.} \quad C(x, \boldsymbol{\theta}[1 : t] \oplus \hat{\boldsymbol{\theta}}^{(t)}) \quad \text{and} \quad x[1 : t-1] = \hat{x}^{(t-1)}[1 : t-1]$$

The Stage $t$ solution, by construction, has $\hat{x}^{(t)}[1:t-1]$ being equal/compatible with the hard commitments from prior stages. The new hard commitments are $\hat{x}_t^{(t)}$, and the rest of the decision vector $\hat{x}^{(t)}[t+1:T]$ are new soft commitments we make for the future stages.

At $t = T$, the prediction $\hat{\theta}^{(T)}$ is a length-0 vector since all the true parameters will have been revealed.

For the rest of the paper, we make the assumption that these Stage $t$ optimizations are *always* satisfiable, regardless of the prior stage solutions, prior+current predictions and revealed parameters. For practical applications, this assumption is both natural and essential. In real-world scenarios, encountering an unsatisfiable condition can lead to catastrophic outcomes. Therefore, before using the application, the domain expert should always have designed the underlying real-world system to have recourse actions to mitigate bad prior commitments (at cost/penalty) and to prevent catastrophe, and furthermore model such recourse actions in the (multi-stage) optimization problem. Any system and the corresponding formulation of multi-stage optimization problem lacking such recourse should not be used/executed. It is thus a reasonable assumption and a practical responsibility our framework asks of the practitioner, that recourse actions are always designed into the underlying system and modelled, so that our feasibility assumption is satisfied.

**Evaluation:** The sequence of predictions $\hat{\theta}^{(0)}, \ldots, \hat{\theta}^{(T)}$, which along with the true parameters $\theta$ induces the sequence of solutions $\hat{x}^{(0)}, \ldots, \hat{x}^{(T)}$, is evaluated using a generalized notion of post-hoc regret, defined as follows (for a minimization problem):

$$PReg((\hat{\theta}^{(0)}, \ldots, \hat{\theta}^{(T)}), \theta) = obj(\hat{x}^{(T)}, \theta) - obj(x^*(\theta), \theta) + \sum_t Pen_t(\hat{x}^{(t-1)} \to \hat{x}^{(t)}, \theta[1:t])$$

where $x^*(\theta)$ is again the optimal in-hindsight vector of decisions for the original Para-OP.

We note that if a problem has only 2 stages (Stages 0 and 1), then our framework of Multi-Stage Predict+Optimize indeed captures Two-Stage Predict+Optimize described in Section 2.

## 4    End-to-End Training Algorithms on MILPs

In this section, we give 3 training algorithms for neural network models for the Multi-Stage Predict+Optimize framework, under the assumption that all Stage $t$ optimization problems can be formulated as (mixed integer) linear programs (MILPs).

The first training algorithm, our baseline (Section 4.1), is a straightforward generalization of the one proposed for the two-stage framework [16]. This algorithm only trains a single neural network and reuses the same parameter predictions across all stages. Although the approach is computationally efficient, it fails to fully exploit the power of the framework, which allows for predictions to be updated at each stage.

Our second and third algorithms (Sections 4.2 and 4.3) instead train one neural network per stage, each making new parameter predictions for the corresponding stage. As mentioned in Section 1, it is delicate to train these neural networks. The quality of a prediction in one stage depends on decisions in other stages, which in turn depends on predictions made in those other stages. To handle this dependency, we employ a coordinate-descent strategy, where each stage/neural network is a coordinate. We present both a sequential version and a parallel version of this strategy as training algorithms, which trade off between training time (sequential being slower) and predictive performance.

We also point out that it is technically possible to train all networks simultaneously, without using coordinate descent like in Sections 4.2 and 4.3. To do so, we would instead use ground truth parameters in place of prior and future stage predictions. However, intuitively, this simpler approach should have worse predictive ability than the proposed two methods, given the interdependency of the predictors. We show experimental comparisons in Appendix H.2 confirming this intuition.

In Section 5, we show empirical results comparing these algorithms and classic non-Predict+Optimize learning algorithms (e.g. standard regression models), which demonstrate that even our baseline training approach outperforms classic non-Predict+Optimize algorithms. Also, our more sophisticated approaches yield even better learning performance, at the cost of additional training time.

## 4.1 Baseline: Extending the Two-Stage Approach to Train a Single Neural Network

We first present a baseline Predict+Optimize training algorithm for our multi-stage setting, via a natural extension of the two-stage approach [16].

This baseline algorithm trains a *single* neural network $NN$, which takes a feature matrix $A$ and returns the prediction $\hat{\boldsymbol{\theta}} = NN(A)$. The same predictions are then reused across all the stages. More specifically, in the language of Section 3, we choose $\hat{\boldsymbol{\theta}}^{(t)} = \hat{\boldsymbol{\theta}}[t+1:T]$ for this basic approach.

We use standard gradient methods to train the neural network $NN$, with the goal of minimizing post-hoc regret as defined in Section 3:

$$PReg = obj(\hat{x}^{(T)}, \boldsymbol{\theta}) - obj(x^*(\boldsymbol{\theta}), \boldsymbol{\theta}) + \sum_t Pen_t(\hat{x}^{(t-1)} \to \hat{x}^{(t)}, \boldsymbol{\theta}[1:t])$$

Noting the second term is independent of $\hat{\boldsymbol{\theta}}^{(0)}$ and hence $NN$, the gradient with respect to an edge weight $w_e$ in $NN$ is

$$\frac{\mathrm{d}\, PReg}{\mathrm{d}w_e} = \frac{\mathrm{d}\, obj(\hat{x}^{(T)}, \boldsymbol{\theta})}{\mathrm{d}w_e} + \sum_{t=1}^{T} \frac{\mathrm{d}\, Pen_t(\hat{x}^{(t-1)} \to \hat{x}^{(t)}, \boldsymbol{\theta}[1:t])}{\mathrm{d}w_e}$$

By the law of total derivative, we can expand this to

$$\frac{\mathrm{d}\, PReg}{\mathrm{d}w_e} = \frac{\mathrm{d}\, obj(\hat{x}^{(T)}, \boldsymbol{\theta})}{\mathrm{d}\hat{x}^{(T)}} \frac{\mathrm{d}\hat{x}^{(T)}}{\mathrm{d}\hat{\boldsymbol{\theta}}} \frac{\mathrm{d}\hat{\boldsymbol{\theta}}}{\mathrm{d}w_e} + \sum_{t=1}^{T} \left( \left. \frac{\partial\, Pen_t}{\partial\hat{x}^{(t-1)}} \right|_{\hat{x}^{(t)}} \frac{\mathrm{d}\hat{x}^{(t-1)}}{\mathrm{d}\hat{\boldsymbol{\theta}}} + \left. \frac{\partial\, Pen_t}{\partial\hat{x}^{(t)}} \right|_{\hat{x}^{(t-1)}} \frac{\mathrm{d}\hat{x}^{(t)}}{\mathrm{d}\hat{\boldsymbol{\theta}}} \right) \frac{\mathrm{d}\hat{\boldsymbol{\theta}}}{\mathrm{d}w_e}$$

The term $\frac{\mathrm{d}\hat{\boldsymbol{\theta}}}{\mathrm{d}w_e}$ is calculated via standard backpropagation, while the terms $\frac{\mathrm{d}\, obj(\hat{x}^{(T)}, \boldsymbol{\theta})}{\mathrm{d}\hat{x}^{(T)}}$, $\left. \frac{\partial\, Pen_t}{\partial\hat{x}^{(t-1)}} \right|_{\hat{x}^{(t)}}$ and $\left. \frac{\partial\, Pen_t}{\partial\hat{x}^{(t)}} \right|_{\hat{x}^{(t-1)}}$ are calculable when the objective and penalty functions are smooth. The only non-trivial calculation is for $\frac{\mathrm{d}\hat{x}^{(t)}}{\mathrm{d}\hat{\boldsymbol{\theta}}}$ for all $t \in [T]$.

Recall that $\hat{x}^{(t)}$ is computed from the Stage $t$ optimization problem, as a deterministic function of $\hat{x}^{(t-1)}$ and $\hat{\boldsymbol{\theta}}^{(t)}$ (which is a sub-vector of $\hat{\boldsymbol{\theta}}$ here, since we reuse predictions), while $\hat{x}^{(t-1)}$ itself also depends on $\hat{\boldsymbol{\theta}}$. We thus further decompose $\frac{\mathrm{d}\hat{x}^{(t)}}{\mathrm{d}\hat{\boldsymbol{\theta}}}$ into the following recursive computation

$$\frac{\mathrm{d}\hat{x}^{(t)}}{\mathrm{d}\hat{\boldsymbol{\theta}}} = \left. \frac{\partial\hat{x}^{(t)}}{\partial\hat{x}^{(t-1)}} \right|_{\hat{\boldsymbol{\theta}}} \frac{\mathrm{d}\hat{x}^{(t-1)}}{\mathrm{d}\hat{\boldsymbol{\theta}}} + \left. \frac{\partial\hat{x}^{(t)}}{\partial\hat{\boldsymbol{\theta}}} \right|_{\hat{x}^{(t-1)}}$$

Calculating $\left. \frac{\partial\hat{x}^{(t)}}{\partial\hat{x}^{(t-1)}} \right|_{\hat{\boldsymbol{\theta}}}$ and $\left. \frac{\partial\hat{x}^{(t)}}{\partial\hat{\boldsymbol{\theta}}} \right|_{\hat{x}^{(t-1)}}$ requires differentiating through a MILP. So instead of directly using MILP formulations for the Stage $t$ optimization problems, we use the convex relaxation by Hu et al. [16], which in turn adapts the approach of Mandi and Guns [22].

We also note that it is possible to use other convex relaxations and approaches to differentiate through the MILP, for example using tools like CvxpyLayers [1]. We chose Hu et al.'s calculations because their experiments showed the computational efficiency of their approach over other tools.

## 4.2 Sequential Coordinate Descent

Training only one neural network, the baseline algorithm is efficient but fails to fully harness the power of the framework in Section 3. Each stage $t$ makes new decisions, and the "goodness" of future decisions depends on these previously committed decisions. Thus, new predictions should be made each stage for future parameters, based on prior-stage optimization decisions, so as to yield current and future optimization decisions that work well with the already-committed ones. However, the baseline algorithm ignores this information and does not update the predictions accordingly.

We thus propose our second training algorithm, which trains one neural network per stage, from Stages 0 to $T-1$. The neural network $NN_t$ for Stage $t$ takes the feature matrix $A$ as input, as well as all the prior decision vector $\hat{x}^{(t-1)}$, and outputs the prediction $\hat{\boldsymbol{\theta}}^{(t)}$ for parameters $\boldsymbol{\theta}[t+1:T]$. The astute reader might recall that the proposed framework in Section 3 allows $NN_t$ to utilize the revealed parameters from stage $t-1$ as additional features as input. However, preliminary

---
**Algorithm 1:** Sequential and Parallel Coordinate Descent Approaches

---

| | |
|---|---|
| **1 PROCEDURE** SeqCoordinateDescent | **6 PROCEDURE** ParCoordinateDescent |
| **2** $t \leftarrow 0$ | **7 While** not terminating |
| **3 While** not terminating | **8**   Save copy $NN'_t \leftarrow NN_t$ for all $t \in [0, T-1]$ |
| **4**   Train $NN_t$ using $NN_{-t}$ (Section 4.2) | **9**   Parallel train+update $NN_t$ using $NN'_{-t}$ (like |
| **5**   $t \leftarrow (t+1) \bmod T$ |   in Section 4.2) |

---

experiments in Appendix H.1 indicated that including such parameters does not really enhance prediction quality, while merely increasing training time. Therefore, in our current implementation, $NN_t$ does not include these revealed parameters. In general, however, such choice should be treated as a hyperparameter and made for each application.

To address the dependency between the neural networks, we employ a coordinate descent approach, where each stage/neural network is treated as a "coordinate". We train the neural networks $NN_0, \ldots, NN_{T-1}$ in cyclic order, repeating until termination (e.g. convergence or timeout). See Algorithm 1 for high-level pseudocode description of this sequential coordinate descent approach, as well as the parallelized version described in the next subsection.

Concretely, we train $NN_t$ by considering all the other neural networks as fixed. Here, we will focus on describing the forward pass, since the backward pass gradient computations follow essentially the same strategy described in Section 4.1 — see Appendix C for details of the gradient computations.

**Forward pass**   Consider a historical (feature, parameter) pair $(A, \boldsymbol{\theta})$. We first iteratively generate the sequence of solutions $\hat{x}^{(0)}, \ldots, \hat{x}^{(t)}$ using $\hat{\boldsymbol{\theta}}^{(i)} = NN_i(A, \hat{x}^{(i-1)})$ for $i \in [0, t-1]$. Then, we compute the Stage $t$ prediction $\hat{\boldsymbol{\theta}}^{(t)} = NN_t(A, \hat{x}^{(t-1)})$, and generate the remaining sequence of solutions $\hat{x}^{(t)}, \ldots, \hat{x}^{(T)}$ using $\hat{\boldsymbol{\theta}}^{(i)} = \hat{\boldsymbol{\theta}}^{(t)}[i+1:T]$ for $i \in [t, T]$.

**Backward pass**   The goal is to compute the derivative of the post-hoc regret with respect to each edge weight $w_e$ of $NN_t$. Similar to Section 4.1, instead of directly using the MILP formulation of all the Stage $t$ optimization problems in the forward pass, we use the convex relaxation of Hu et al. and Mandi and Guns. This allows us to differentiate the modified (due to convex relaxations) post-hoc regret with respect to each $w_e$ in $NN_t$. As mentioned, the calculations are quite similar to those in Section 4.1, and so we defer them to Appendix C.

Experimental results in Section 5 demonstrate that the sequential coordinate descent training approach outperforms both the classic non-Predict+Optimize methods and the baseline Predict+Optimize approach from Section 4.1.

We note that in the above description of the training implementation of $NN_t$, there is a lot of repeated computation that can be pre-computed and reused. Since, during coordinate descent, the prior neural networks $NN_0, \ldots, NN_{t-1}$ are considered fixed, the solutions $(\hat{x}^{(0)}, \ldots, \hat{x}^{(t-1)})$ are also fixed for a given (features, parameters) pair $(A, \boldsymbol{\theta})$ *no matter* how $NN_t$ is updated during training. Thus, for each $(A, \boldsymbol{\theta})$ we always pre-compute and save the sequence of solutions $(\hat{x}^{(0)}, \ldots, \hat{x}^{(t-1)})$, and only recompute $(\hat{x}^t, \ldots, \hat{x}^T)$ as we update $NN_t$ through training gradient steps.

## 4.3   Parallel Coordinate Descent

While the sequential coordinate descent training approach yields accurate predictors, the computational cost is also high. Training a single neural network already requires solving sequences of optimization problems over many iterations; the serialization from training the neural networks one at a time can make the resulting training time prohibitive for applications.

We thus propose to parallelize the coordinate descent approach, slightly sacrificing prediction quality for efficiency. In each coordinate descent step, we train all the neural networks in parallel. When training a particular neural network $NN_t$, we use copies of $NN_0, \ldots, NN_{t-1}, NN_{t+1}, \ldots, NN_{T-1}$ from the previous descent step, but otherwise the training implementation remain the same as in Section 4.2. See also Algorithm 1.

This simple change drastically improves running time (Appendix G), while only slightly decreases predictive accuracy: the post-hoc regret of models trained using parallel coordinate descent sits between that of the baseline training algorithm and the sequential coordinate descent approach.

## 5 Experimental Evaluation

We evaluate the proposed 3 training algorithms: Baseline, Sequential Coordinate Descent (SCD), and Parallel Coordinate Descent (PCD) on 3 benchmarks described in Appendix D. We compare these algorithms to classic non-Predict+Optimize regression models [10]: ridge regression (Ridge), $k$-nearest neighbors ($k$-NN), classification and regression tree (CART), random forest (RF), and neural network (NN). The single predictions from these classical regression models are used in test time identically to our Baseline approach (Section 4.1). We tune all algorithm hyperparameters via cross-validation — Appendix E gives all the hyperparameter types and chosen values. In particular, the termination criteria for SCD and PCD in Algorithm 1 are based on a threshold that measures the difference in training set post-hoc regrets between two (outermost) iterations of the training coordinate descent. This threshold is also treated as a hyperparameter that was tuned per each application; in the experiments presented in this paper, we used a threshold of 0.1.

Due to space limitations, we present only the best results obtained among all standard regression methods (BAS) as one column in the main paper. See Appendix F for full results. Furthermore, we report mainly the prediction accuracy — see Appendix G for computational setup and detailed runtime comparisons.

Given the lack of datasets specific to these benchmarks, we follow a standard Predict+Optimize experimental approach [16, 24, 7] and use real data from different problems as numerical values in our experiment instances. We include details in the individual subsections. For each benchmark, we run 30 simulations, each simulation containing a 70/30 training/test data split.

### 5.1 Production and Sales Problem

Our first benchmark is a linear programming (LP) problem. An oil company is developing a production and sales plan for the upcoming $T$ quarters/months. The company aims to maximize profits — sales revenues minus production costs — under the constraint that the amount of oil sold each quarter/month cannot exceed the customer demands. The production cost and sales price for each quarter/month are known, but the demand is revealed only at the beginning of each quarter/month after the company receives the orders. See Appendix D.1 for the detailed description and LP model.

We generate production costs and sales prices following the method by Ardjmand et al. [3]. Production costs are randomly generated from $[50, 100]$. Two groups of sales prices are considered: low-profit product prices are randomly generated from $[50, 100]$; high-profit product prices are from $[120, 150]$. Customer demands are the unknown parameters and need prediction — we use real data from a knapsack benchmark [28] as demand parameters, where each parameter is related to 4096 features.

We conduct experiments on $T = \{4, 12\}$, corresponding to 4 quarters or 12 months. For NN, Baseline, SCD, and PCD, we use 5-layer fully connected networks with 512 neurons per hidden layer.

Table 1 reports the mean post-hoc regrets and standard deviations across 30 simulations for the proposed 3 methods and BAS on the problem. Appendix F.1 gives a full data table (Table 5) with all standard regression methods. We also report the mean and standard deviations of True Optimal Values (TOV) in the last column — optimal objective under true parameters in hindsight — for readers to use as (rough) normalization for relative errors. The results demonstrate the advantage of Predict+Optimize methods. All three proposed methods, even Baseline, beat all standard regression methods. SCD consistently achieves the best performance, followed closely by PCD. Baseline falls between the two coordinate descent methods.

In Appendix F.1, Table 6 shows the percentage improvements of the proposed 3 methods against BAS. From that table, we observe that the advantage of our methods increases with the number of stages.

Considering the relatively large standard deviations in Table 1, to show the substantial performance improvements clearly, we provide "win rate" results in Figure 1 and related information in Table 7 in Appendix F.1. Here, "win rate" refers to counting the number of simulations (out of 30) where a

Table 1: Mean post-hoc regrets and standard deviations for the production and sales problem.

| Price group | Stage num | SCD | PCD | Baseline | BAS | TOV |
|---|---|---|---|---|---|---|
| Low-profit | 4 | **293.78±99.21** | 297.34±107.44 | 305.26±100.88 | 355.56±103.78 | 1615.75±675.77 |
| | 12 | **488.72±127.62** | 495.21±122.42 | 515.80±137.67 | 637.77±199.25 | 7344.78±2290.04 |
| High-profit | 4 | **505.24±89.55** | 520.76±92.20 | 526.77±104.99 | 561.36±96.49 | 5007.09±976.65 |
| | 12 | **887.38±250.55** | 905.61±255.99 | 935.03±263.47 | 997.44±273.91 | 21066.00±4159.56 |

Table 2: Mean post-hoc regrets and standard deviations for the investment problem when capital=50.

| Stage num | Transaction factor | SCD | PCD | Baseline | BAS | TOV |
|---|---|---|---|---|---|---|
| 4 | 0.01 | **47.48±6.98** | 47.67±6.64 | 48.24±7.13 | 48.98±7.19 | 158.56±11.19 |
| | 0.05 | **34.92±5.57** | 35.22±5.98 | 35.64±6.28 | 36.42±5.70 | 126.22±8.86 |
| | 0.1 | **25.50±3.88** | 25.63±3.93 | 25.96±4.64 | 26.62±4.07 | 99.83±7.02 |
| 12 | 0.01 | **3846.20±420.94** | 3869.76±420.01 | 3941.09±437.57 | 3999.70±475.44 | 6166.73±573.51 |
| | 0.05 | **1663.82±208.60** | 1679.17±205.01 | 1701.51±222.45 | 1736.59±232.15 | 2860.05±288.85 |
| | 0.1 | **646.14±75.52** | 652.57±74.45 | 665.71±76.40 | 680.94±70.76 | 1259.99±107.60 |

method outperforms another. Table 7 reports win rate results among Baseline, SCD, PCD, and BAS, and Figure 1 further compares the proposed methods against BAS across each individual simulation. Both the table and figure show that SCD outperforms BAS in all simulations, while PCD and Baseline outperform BAS in most simulations.

## 5.2 Investment Problem

Our second benchmark shows our framework on a MILP problem. The task is to create an investment plan for trading financial products over a year, under limited capital. The net profit consists of: 1) dividends gained from holding products, 2) market price of products in the portfolio at the end of the year and 3) net profits from trading products (selling price minus buying price) minus trading transaction fees. For simplicity, we assume that transaction fees are linear in the products' market prices. We call the proportionality constant the *transaction factor*, which we vary across experiments.

The market price of (*resp.* dividends from owning) each product in each quarter/month is revealed only at the beginning (*resp.* end) of the quarter/month, requiring predictions. As with the previous benchmark, we use real data from a knapsack benchmark [28] for parameters needing prediction. The detailed problem description and the MILP formulation are in Appendix D.2

We conduct experiments on five financial products, and $T = \{4, 12\}$, corresponding to 4 quarters or 12 months. We consider two different initial capital values: 25 and 50, and three different transaction factors: 0.01, 0.05, or 0.1. For NN, Baseline, SCD and PCD, we again use a 5-layer fully connected network with 512 neurons per hidden layer.

Table 2 reports the mean post-hoc regrets and standard deviations across 30 simulations for the proposed three methods and BAS on the problem when capital=50. We report full data tables (Tables 8 and 9) in Appendix F.2. TOVs are reported in the last column, again for reference. The results confirm the performance ordering of SCD > PCD > Baseline > classical regression methods.

Table 10 in Appendix F.2 presents the percentage improvement of our proposed methods against BAS. Curiously, contrasting the previous benchmark, the advantage of our methods against BAS appears to be non-increasing with the number of stages. We hypothesize this is an LP versus MILP issue — training for this benchmark required relaxing integrality constraints, introducing more approximations with more stages. It is interesting future work to study if the performance decay in the number of stages is a general phenomenon for MILPs and if it can be mitigated.

For the investment problem benchmark, we again provide win rate results in Table 11 in Appendix F.2. The results show that the proposed 3 methods outperform BAS in more than 96.67% of the simulations.

## 5.3 Nurse Rostering Problem

Our last benchmark is the nurse rostering MILP problem described in Section 1. The task is to minimize the costs of hiring nurses to meet the (unknown) patient demands. After each stage's patient

Table 3: Mean post-hoc regrets and standard deviations for the nurse rostering problem.

| Extra nurse payment | SCD | PCD | Baseline | BAS | TOV |
|---|---|---|---|---|---|
| 15 | **607.66±142.19** | 622.05±153.64 | 629.35±153.67 | 662.34±169.17 | 10611.64±1574.11 |
| 20 | **789.65±200.22** | 805.11±224.99 | 817.60±219.47 | 863.02±214.50 | 10732.32±1504.12 |
| 25 | **1038.29±255.42** | 1048.08±281.32 | 1083.45±259.68 | 1144.63±305.00 | 10893.54±1485.37 |
| 30 | **1207.50±319.25** | 1240.48±332.39 | 1290.10±371.08 | 1369.81±373.20 | 11110.73±1344.15 |

demands are revealed, the admin can modify future rosters at cost, and hire extra temporary nurses at higher salary in case of understaffing. See Appendix B for a detailed description of the model.

Each problem instance consists of 10 regular nurses and 7 days (stages). Extra nurses come at a cost of $\{15, 20, 25, 30\}$ in different experiments. Due to the longer solving time for these MILPs, we use real data from the ICON scheduling competition [32] as the numerical values for patient demands, where each demand value is related to 8 features as opposed to 4096 features in previous benchmarks. Given far fewer features, for both NN, Baseline, SCD and PCD, we use a smaller network architecture: a 5-layer fully-connected network with 16 neurons per hidden layer.

Table 3 reports the mean post-hoc regrets and standard deviations across 30 simulations for each approach, again corroborating the prediction performance order of SCD > PCD > Baseline > BAS. Appendix F.3 gives full experimental results.

Similarly to the first two benchmarks, we report win rate results in Table 14 and show comparisons between the proposed methods against BAS across each individual simulation in Figure 2 in Appendix F.3. Table 14 shows that SCD pretty consistently outperforms BAS, achieving win rates of 86.67% or higher in most scenarios. PCD also demonstrates competitive performance, with win rates ranging from 70% to 83.33%, making it a viable alternative as we hypothesized.

## 5.4   Runtime Analysis

Appendix G gives the training times for each method. The training times follow the order of SCD > PCD > Baseline > classic regression methods, which is the same for predictive accuracy in the benchmarks, indicating a tradeoff between training time and accuracy. Our methods take longer runtime than most other regression methods due to having to solve sequences of linear programs during training. Among our methods, the coordinate descent methods take longer time than Baseline due to having to train more neural networks, and the sequential version naturally takes longer than the parallelized version.

## 6   Conclusion and Future Work

We propose the first Predict+Optimize framework for scenarios where unknown parameters are revealed progressively over stages. Specifically, our proposal allows better predictions and (re)optimization at each stage as more parameters are made known. Algorithmically, we focus on MILPs——a large and widely-studied class of problems——and present three training methods for our novel framework. Empirical results in three benchmarks demonstrate better predictions from our methods over classical ones. Our methods trade off between predictive accuracy and training time.

Our work establishes the feasibility of Multi-Stage Predict+Optimize, and furthermore shows that even our baseline algorithm of training a single predictor outperforms classical non-Predict+Optimize approaches. Looking into the future beyond the present work, there is ample space for algorithmic improvements. As we observed in Section 5, the current experimental results suggest that our multi-stage predict+optimize methods display rather different behaviors depending on whether the optimization is a linear program or a mixed integer program. In particular, for linear programs, the advantage of our methods over classical methods *increases* with the number of stages, whereas the opposite happens for integer programs. We believe it is important to investigate whether this phenomenon holds more generally. If so, it is a prudent research direction to understand whether such decay is inevitable for MILPs, or if there are algorithmic techniques to mitigate this effect.

## Acknowledgments

We thank the anonymous reviewers for their constructive comments. In addition, Xinyi Hu and Jimmy H.M. Lee acknowledge the financial support of a General Research Fund (RGC Ref. No. CUHK 14206321) by the Hong Kong University Grants Committee, and also a Direct Grant (Ref. No. 4055231) by The Chinese University of Hong Kong. Jasper C.H. Lee's work was partially done while he was at the University of Wisconsin-Madison, supported by a Croucher Fellowship for Postdoctoral Research and NSF Medium Award CCF-2107079. Peter Stuckey's contribution was partially supported by the DARPA Assured Neuro Symbolic Learning and Reasoning (ANSR) program under award number FA8750-23-2-1016, and by the Australian Research Council through the OPTIMA ITTC IC200100009.

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

# A Detailed Literature Review

In this section, we first summarize the related works in Predict+Optimize, and then summarize other works related to learning unknowns in optimization problems, but outside of Predict+Optimize.

## A.1 Predict+Optimize

As mentioned in Section 1, prior works all focus on the case where all unknown parameters are revealed simultaneously. Most of them have focused on the regime where the unknown parameters only appear in the objective and use the regret proposed by Elmachtoub et al. [9] as the loss function. Since the regret loss is usually not (sub-) differentiable, and gradient-based methods do not apply, they proposed ways to overcome the non-differentiability of the regret. Elmachtoub et al. [9] propose a differentiable surrogate function for the regret function, while Wilder et al. [36] relax the integral objective in constrained optimization and solve a regularized quadratic programming problem. Mandy and Guns [22] focus on mixed integer linear programs and propose an interior point based approach. In addition to computing the (approximate) gradients of the regret function or approximations of it, another way to deal with the non-differentiability of the regret is to exploit the structure of optimization problems to train models without computing gradients. Demirović et al. [7] investigate problems amenable to tabular dynamic programming and propose a coordinate descent method to learn a linear prediction function. Hu et al. [17] extend their framework, to enable problems solvable with a recursive or iterative algorithm to be tackled in Predict+Optimize. Guler et al. [12] proposes a divide and conquer algorithm, extending the work of Demirović et al. [7] in a different manner so that the algorithm can deal with problems with the linear objective function.

As for Predict+Optimize with unknown parameters also in constraints, Hu et al. [15] first propose the post-hoc regret loss function and a framework for packing and covering LPs with unknown parameters in both objectives and constraints. They [16] further advocate a conceptually simpler framework, which enable solving MILPs with unknown constraints. Besides, there are also works applying Predict+Optimize to a wide range of real-world problems, including maritime transportation [35], last-mile delivery [6], and trading in renewable energy [34].

## A.2 Decision-Focused Learning

Now we summarize other works related to learning unknowns in optimization problems, particularly those outside of Predict+Optimize. These works can be placed into two categories.

One category considers learning unknown parameters but with very different goals and measures of loss. For example, CombOptNet [28] and Nandwani et al. [25] focus on learning parameters so as to make the predicted optimal solution (Stage 0 estimated solution in the proposed framework) as close to the true optimal solution $x^*$ as possible in the solution space/metric. However, these works also assume that all unknown parameters are revealed simultaneously, and thus cannot be applied to applications where unknown parameters are revealed progressively over several stages. Furthermore, experiments in Two-Stage Predict+Optimize [16] show that these other methods yield worse predictive performance when evaluated on the post-hoc regret.

Another category gives ways to differentiate through LPs or LPs with regularizations, as a technical component in a gradient-based training algorithm [2, 36, 1]. While our proposed algorithms in Section 4.1 and Appendix C use the methods of Hu et al. [15, 16] and Mandi and Guns [22] to perform gradient computations, we could in principle use any of the aforementioned other works. However, we point out that our main contribution is not the gradient computation method but the two training algorithms of the set of NNs. Nonetheless, experiments in Two-Stage Predict+Optimize framework [16] demonstrate that the gradient calculation method they used (which we also use) performs at least as well in post-hoc regret performance as other gradient methods, while being (significantly) faster. This is the reason we follow Hu et al.'s method and implementation.

## A.3 (Multi-Stage) Stochastic Programming

As mentioned in Section 1, while stochastic programming and Predict+Optimize are related frameworks, the technical challenges are very different. The most important difference is that Predict+Optimize is a supervised learning problem, whereas stochastic programming is unsupervised

learning. In Predict+Optimize frameworks, the true parameters (which need prediction) are always associated with relevant features that help prediction. On the other hand, stochastic programming frameworks have no such features, and typically assume that the entire distribution over the unknown parameters is given to the algorithm — in practice, the distribution needs to be estimated from historical data over the unknown parameters, which is an unsupervised density estimation problem.

Due to the different starting assumptions, Predict+Optimize and stochastic programming formulate optimization problems rather differently. In stochastic programming, since the assumption is that the full parameter distribution is given, the optimization problem (or problems, across stages) would explicitly include the expectation operator in the objective — the goal is to solve for optimization decisions so that the expected objective, with expectation taken over the parameter distribution, is maximized/minimized. Predict+Optimize frameworks approach this rather differently: while the goal is still to optimize the expected objective, the optimization problems themselves are phrased such that they take predicted parameters, and the problem asks for the optimal decisions assuming the predicted parameters. It then becomes the goal of the *learning algorithm* to learn to make predictions from features, such that the expected objective is optimized overall. This is achieved via empirical risk minimization over training data, which we assume are samples from the underlying (feature,parameter) joint distribution.

We also note the dimensionality of the objects being learnt in the different frameworks. In stochastic programming, the entire distribution over the unknown parameters needs to be learnt. On the other hand, in Predict+Optimize, we learn a mapping from features to predicted parameters, which, under smoothness assumptions or bounded model complexity assumptions (e.g. by restricting to using a fixed neural network architecture), can effectively be regarded as a (much) lower dimensional object than learning an entire distribution over unknown parameters.

### A.4 Integration of Machine Learning and Mixed-Integer Programming

The integration of machine learning and (discrete) optimization is an increasingly popular field. In this section, we mention some works on using machine learning to help solve mixed-integer programs and especially stochastic programs.

In the context of integrating machine learning with stochastic programming, several noteworthy contributions have emerged. Donti et al. [8] propose an end-to-end learning framework that directly aligns the training of probabilistic machine learning models with the ultimate task-based objective in the context of stochastic programming. Patel et al. [27] use a neural network to approximate the expected value function in two-stage stochastic programming problems, enabling more efficient solution approaches compared to traditional methods. Bae et al. [4] propose a neural network-based stagewise decomposition algorithm that can effectively approximate value functions for large-scale multistage stochastic programming problems. Rychener et al. [31] develop a principled end-to-end learning framework for training neural network decision maps that can effectively solve stochastic optimization problems, including empirical risk minimization and distributionally robust optimization formulations.

The application of using ML to help mixed-integer programming (MIP) has also seen substantial progress: ML algorithms for exact solving of MIPs by branch-and-cut based algorithms [5, 11, 37], ML algorithms for exact solving of MIPs by decomposition-based algorithms [19, 21], ML algorithms for approximate solving MIPs by large neighborhood search based algorithm [33, 20], and so on.

## B A Detailed Example for Multi-Stage Predict+Optimize Framework

In this section, we use the hospital scenario, i.e., the nurse rostering problem (NRP), mentioned in Section 1 as a running example for the Multi-Stage Predict+Optimize framework described in Section 3.1.

Here we describe the NRP in detail. A hospital needs to make nurses schedule for the whole week (7 days) two weeks beforehand so that the nurses can be well prepared for the work and also plan for their leisure activities. The goal of the hospital is to minimize the total costs for hiring nurses and meet the patients' demands.

There are full-time nurses in the hospital. If there are too many patients and the hospital's nurses are understaffed, the hospital can temporarily hire some extra nurses at a higher salary. Since the number of patients that will come in each shift on each day is unknown two weeks beforehand, the hospital needs to predict the number of patients to make a schedule for the full-time nurses and plan to hire extra nurses. The hospital will learn the predictor based on historical hospital records, considering features such as time of year, day of the week and temperature.

To provide better service to patients, the hospital has an appointment system that requires patients to schedule an appointment in advance to receive medical care. Reservations for the next day close the night before. At this point, the hospital knows the precise number of patients for each shift of the current day. Therefore, at the night of day $(t-1)$, i.e., Stage $t$ $(1 \le t \le 7)$, the true numbers of patients for each shift of the current day are revealed.

Now we show the running example for the Multi-Stage Predict+Optimize framework. Examples 1, and 2 are examples for Stage 0 and Stage $t$ (for $1 \le t \le T$) respectively.

**Example 1.** *Suppose there are $n$ full-time nurses, 7 days, and 3 working shifts per day. Full-time nurses are entitled to take a rest: day-off shift. The decision variables are: 1) a Boolean vector $x \in \{0,1\}^{n \times 7 \times 3}$, where $x_{i,j,k}$ represents that whether nurse $i$ is assigned to shift $k$ in day $j$, and 2) an integer vector $\sigma \in \mathbb{N}^{7 \times 3}$, where $\sigma_{j,k}$ represents the number of extra nurses hired in shift $k$ day $j$. Let $d_{j,k}$ denote the number of patients in shift $k$ day $j$, $m_i$ denote the number of patients that the nurse $i$ can serve per shift, $c_i$ denote the payment of the nurse $i$ per shift, $e_s$ denote the number of patients that each extra nurse can serve per shift, and $e_c$ denote the payment of each extra nurse per shift. The unknown parameters are $\boldsymbol{d} \in \mathbb{N}^{7 \times 3}$.*

*Consider the time that the schedules need to be made as Stage 0. The hospital learns the predictor and uses the estimated number of patients $\hat{\boldsymbol{d}}^{(0)}$ to optimize for that week's schedule. The Stage 0 OP, the NRP using the estimations, can be formulated as:*

$$\hat{x}^{(0)}, \hat{\sigma}^{(0)} = \arg\min_{x,\sigma} \sum_{i=1}^{n} c_i \sum_{j=1}^{7} \sum_{k=1}^{3} x_{i,j,k} + e_c \sum_{j=1}^{7} \sum_{k=1}^{3} \sigma_{j,k} \qquad (1)$$

$$s.t. \sum_{i=1}^{n} m_i x_{i,j,k} + e_s \sigma_{j,k} \ge \hat{d}_{j,k}^{(0)}, \quad \forall j \in \{1,\ldots,7\}, k \in \{1,2,3\} \qquad (2)$$

$$\sum_{k=1}^{4} x_{i,j,k} = 1, \quad \forall i \in \{1,\ldots,n\}, j \in \{1,\ldots,7\} \qquad (3)$$

$$x_{i,j,3} + x_{i,j+1,1} \le 1, \quad \forall i \in \{1,\ldots,n\}, j \in \{1,\ldots,6\} \qquad (4)$$

$$1 \le \sum_{j=1}^{7} x_{i,j,4} \le 2, \quad \forall i \in \{1,\ldots,n\} \qquad (5)$$

$$x \in \{0,1\}, \quad \sigma \ge 0 \qquad (6)$$

*where Equation (1) represents the objective, which is to minimize the total costs for hiring full-time nurses and extra nurses; Equation (2) ensures that the schedule can satisfy the patient demand under each shift; Equation (3) ensures that each full-time nurse will be scheduled for exactly one shift each day; Equation (4) ensures that no full-time nurse will be scheduled to work a night shift followed immediately by a morning shift; and Equation (5) ensures that each full-time nurse gets one or two day-off shifts per week.*

*After Stage 0, the schedules for day 1 are hard commitments and cannot be changed, i.e., $\hat{x}_0^{(0)} = \{x_{i,1,k} \mid \forall i \in \{1,...,n\}, k \in \{1,2,3\}\}$, whereas the rest of the decisions are soft commitments.*

**Example 2.** *(**Continued**) At the night of day $t-1$, i.e., Stage $t$ (for $1 \le t \le 7$), the reservations for the next day close, and the true numbers of patients for the three shifts of the next day $\theta_t = (d_{t,1}, d_{t,2}, d_{t,3}) \in \mathbb{N}^3$ are revealed. By Stage $t$, all the true numbers of patients for the prior $t-1$ days are also revealed. The number of patients for the later $7-t$ days are still uncovered and are estimated as $\hat{\theta}^{(t)} = (\hat{\theta}_{t+1}^{(t)}, \ldots, \hat{\theta}_T^{(t)})$, where $\hat{\theta}_i^{(t)} = (\hat{d}_{i,1}^{(t)}, \hat{d}_{i,2}^{(t)}, \hat{d}_{i,3}^{(t)}) \in \mathbb{N}^3$ represents the numbers of patients on day $i$ estimated on day $t$.*

*Hard commitments contain two parts: 1) the schedule for the day $t$ and the prior $t-1$ days, and 2) the number of extra nurses hired in the prior $t-1$ days, i.e., here $x[1:t-1]$ represents $\{x_{i,j,k} \mid \forall i \in$*

$\{1, ..., n\}, j \in \{1, ..., t\}, k \in \{1, 2, 3\}\} \cup \{\sigma_{j,k} \mid \forall j \in \{1, ..., t-1\}, k \in \{1, 2, 3\}\}$. *The hospital may update the predictions and reschedule for the later $(7-t)$ days. But such rescheduled leads to extra costs for hiring full-time nurses, which are recorded by the penalty function $Pen(\hat{x}^{(t-1)} \to x, \boldsymbol{\theta}[1:t])$. The more temporarily the shift is rescheduled, the larger the increase in the costs. For simplicity, we assume that the extra cost is linear in the original cost for hiring each full-time nurse. In this scenario, the penalty function can be formulated as $Extra(\hat{x}^{(t-1)} \to x)$:*

$$Extra(\hat{x}^{(t-1)} \to x) = \sum_{i=1}^{n} \sum_{j=1}^{7} \sum_{k=1}^{3} Extra(\hat{x}^{(t-1)} \to x)_{i,j,k}$$

*where the $(i, j, k)$-th item in the penalty function is:*

$$Extra(\hat{x}^{(t-1)} \to x)_{i,j,k} = \begin{cases} (T-j+t)\rho_i c_i, & x_{i,j,k} > \hat{x}_{i,j,k}^{(t-1)} \\ 0, & x_{i,j,k} \le \hat{x}_{i,j,k}^{(t-1)} \end{cases}$$

*As mentioned in Section 3.1, the Stage $t$ optimization problem modifies the original Para-OP by: 1) introducing the penalty term capturing the cost of changing the Stage $t-1$ solution $\hat{x}^{(t-1)}$ to the new Stage $t$ solution $x$ to the objective:*

$$\hat{x}^{(t)}, \hat{\sigma}^{(t)} = \arg\min_{x,\sigma} \sum_{i=1}^{n} c_i \sum_{j=1}^{7} \sum_{k=1}^{3} x_{i,j,k} + e_c \sum_{j=1}^{7} \sum_{k=1}^{3} \sigma_{j,k} + \sum_{i=1}^{n} \sum_{j=1}^{7} \sum_{k=1}^{3} Extra(\hat{x}^{(t-1)} \to x)_{i,j,k}$$

*and 2) introducing the constraint that hard commitments from prior stages cannot be modified in the current Stage $t$:*

$$x_{i,j,k} = \hat{x}_{i,j,k}^{(t-1)}, \quad \forall i \in \{1, ..., n\}, j \in \{1, ..., t\}, k \in \{1, 2, 3\}\}$$
$$\sigma_{j,k} = \hat{\sigma}_{j,k}^{(t-1)}, \quad \forall j \in \{1, ..., t-1\}, k \in \{1, 2, 3\}$$

*Besides, for the first constraint in Equation (2), the Stage 0 predicted parameters $\hat{d}^0$ are replaced by $(d_{1,1}, \ldots, d_{t,3}, \hat{d}_{t+1,1}^{(t)}, \ldots, \hat{d}_{7,3}^{(t)})$:*

$$\sum_{i=1}^{n} m_i x_{i,j,k} + e_s \sigma_{j,k} \ge d_{j,k}, \quad \forall j \in \{1, \ldots, t\}, k \in \{1, 2, 3\}$$
$$\sum_{i=1}^{n} m_i x_{i,j,k} + e_s \sigma_{j,k} \ge \hat{d}_{j,k}^t, \quad \forall j \in \{t+1, \ldots, 7\}, k \in \{1, 2, 3\}$$

*The four constraints from Equation (3) to Equation (6) keep the same in the Stage $t$ (for $1 \le t \le 7$) optimization.*

## C  Gradient Computations in Sequential Coordinate Descent

The post-hoc regret used to train $NN_t$ can be written as:

$$PReg(\hat{\boldsymbol{\theta}}^{(t)}, \boldsymbol{\theta}[t+1:T]) = obj(\hat{x}^{(T)}, \boldsymbol{\theta}) - obj(x^*(\boldsymbol{\theta}), \boldsymbol{\theta}) + \sum_{i=t}^{T} Pen_i(\hat{x}^{(i-1)} \to \hat{x}^{(i)}, \boldsymbol{\theta}[1:i]) \quad (7)$$

Noting the second term is independent of $\hat{\boldsymbol{\theta}}^{(t)}$ and hence $NN_t$, the gradient with respect to an edge weight $w_e$ in $NN_t$ is

$$\frac{\mathrm{d}\, PReg}{\mathrm{d}w_e} = \frac{\mathrm{d}\, obj(\hat{x}^{(T)}, \boldsymbol{\theta})}{\mathrm{d}w_e} + \sum_{i=t}^{T} \frac{\mathrm{d}\, Pen_i(\hat{x}^{(i-1)} \to \hat{x}^{(i)}, \boldsymbol{\theta}[1:i])}{\mathrm{d}w_e}$$

By the law of total derivative, we can expand this to

$$\frac{\mathrm{d}\, PReg}{\mathrm{d}w_e} = \frac{\mathrm{d}\, obj(\hat{x}^{(T)}, \boldsymbol{\theta})}{\mathrm{d}\hat{x}^{(T)}} \frac{\mathrm{d}\hat{x}^{(T)}}{\mathrm{d}\hat{\boldsymbol{\theta}}^{(t)}} \frac{\mathrm{d}\hat{\boldsymbol{\theta}}^{(t)}}{\mathrm{d}w_e} + \sum_{i=t}^{T} \left( \frac{\partial\, Pen_i}{\partial \hat{x}^{(i-1)}} \bigg|_{\hat{x}^{(i)}} \frac{\mathrm{d}\hat{x}^{(i-1)}}{\mathrm{d}\hat{\boldsymbol{\theta}}^{(t)}} + \frac{\partial\, Pen_i}{\partial \hat{x}^{(i)}} \bigg|_{\hat{x}^{(i-1)}} \frac{\mathrm{d}\hat{x}^{(i)}}{\mathrm{d}\hat{\boldsymbol{\theta}}^{(t)}} \right) \frac{\mathrm{d}\hat{\boldsymbol{\theta}}^{(t)}}{\mathrm{d}w_e}$$

Similar to the gradient computation in Section 4.1, the term $\frac{\mathrm{d}\hat{\boldsymbol{\theta}}^{(t)}}{\mathrm{d}w_e}$ is calculated via standard backpropagation, while the terms $\frac{\mathrm{d}\,obj(\hat{x}^{(T)},\boldsymbol{\theta})}{\mathrm{d}\hat{x}^{(T)}}$, $\frac{\partial\,Pen_i}{\partial\hat{x}^{(i-1)}}\big|_{\hat{x}^{(i)}}$ and $\frac{\partial\,Pen_i}{\partial\hat{x}^{(i)}}\big|_{\hat{x}^{(i-1)}}$ are calculable when the objective and penalty functions are smooth. The only non-trivial calculation is for $\frac{\mathrm{d}\hat{x}^{(i)}}{\mathrm{d}\hat{\boldsymbol{\theta}}^{(t)}}$ for all $i \in [t:T]$.

Recall that $\hat{x}^{(i)}$ is computed from the Stage $i$ optimization problem, as a deterministic function of $\hat{x}^{(i-1)}$ and $\hat{\boldsymbol{\theta}}^{(t)}$, while $\hat{x}^{(i-1)}$ itself also depends on $\hat{\boldsymbol{\theta}}^{(t)}$. We thus further decompose $\frac{\mathrm{d}\hat{x}^{(i)}}{\mathrm{d}\hat{\boldsymbol{\theta}}^{(t)}}$ into the following recursive computation

$$\frac{\mathrm{d}\hat{x}^{(i)}}{\mathrm{d}\hat{\boldsymbol{\theta}}^{(t)}} = \frac{\partial\hat{x}^{(i)}}{\partial\hat{x}^{(i-1)}}\bigg|_{\hat{\boldsymbol{\theta}}^{(t)}}\frac{\mathrm{d}\hat{x}^{(i-1)}}{\mathrm{d}\hat{\boldsymbol{\theta}}^{(t)}} + \frac{\partial\hat{x}^{(i)}}{\partial\hat{\boldsymbol{\theta}}^{(t)}}\bigg|_{\hat{x}^{(i-1)}}$$

Calculating $\frac{\partial\hat{x}^{(i)}}{\partial\hat{x}^{(i-1)}}\big|_{\hat{\boldsymbol{\theta}}^{(t)}}$ and $\frac{\partial\hat{x}^{(i)}}{\partial\hat{\boldsymbol{\theta}}^{(t)}}\big|_{\hat{x}^{(i-1)}}$ requires differentiating through a MILP. So instead of directly using MILP formulations for the Stage $t$ optimization problems, we use the convex relaxation by Hu et al. [16], which in turn adapts the approach of Mandi and Guns [22].

## D  Details for Case Studies

In this section, we give a detailed description for the other two benchmarks used in Section 5.

### D.1  Production and Sales Problem

We first demonstrate, using the example of the production and sales problem, how our framework can tackle LPs. An oil company intends to develop a production and sales plan for the upcoming $T$ quarters/months. The goal is to maximize the profits, i.e., the sales revenues minus the production costs, under the constraints that the amount of oil product sold each quarter/month cannot exceed the customer demands. The production cost and sales price for each quarter/month are known, but the customer demand is revealed only at the beginning of each quarter/month after the company receives the orders. The company will estimate the customer demands based on historical sales records, considering features such as oil type, oil consumption of different areas, and so on.

The decision variables are: 1) a real vector $x \in \mathbb{R}^T$, where $x_i$ represents the amount of product produced in month $i$, and 2) a real vector $y \in \mathbb{R}^T$, where $y_i$ represents the amount of product sold in month $i$. Let $p_i$ denote the unit profit of selling product in month $i$, $c_i$ denote the unit cost of producing product in month $i$, $d_i$ denote the customer demands in month $i$. The unknown parameters are $\boldsymbol{d} \in \mathbb{R}^T$.

At Stage 0, i.e., the time that the production and sales plan needs to be made, there is no order yet and the customer demands are unknown. The company learns the predictor and uses the predicted demands $\hat{\boldsymbol{d}}^{(0)}$ to make the plan. The Stage 0 OP can be formulated as:

$$\hat{x}^{(0)}, \hat{y}^{(0)} = \arg\max_{x,y} \sum_{i=1}^{T} p_i y_i - \sum_{i=1}^{T} c_i x_i \tag{8}$$

$$\text{s.t. } y_i \leq \hat{d}_i^{(0)}, \quad \forall i \in \{1, \ldots, T\} \tag{9}$$

$$y_i \leq \sum_{j=1}^{i-1} x_j - \sum_{j=1}^{i-1} y_j, \quad \forall i \in \{1, \ldots, T\} \tag{10}$$

$$x \geq 0, \quad y \geq 0 \tag{11}$$

where Equation (8) represents the objective, for maximizing the profits, i.e., the sales revenues minus the production costs; Equation (9) ensures that the amount of oil product sold each quarter/month will not exceed the customer demands; Equation (10) ensures that the amount of oil product sold each quarter/month will not exceed the inventory at that quarter/month.

At the beginning of each quarter/month, the company receives orders, and the demand for that quarter/month is revealed. We assume that the beginning of each quarter/month is one stage. Then, by Stage $t$ $(1 \leq t \leq T)$, all the true demands for the prior $(t-1)$ quarters/months as well as the $t$

quarter/month are revealed. The demands for the later $(T - t)$ quarters/months are still uncovered and are estimated as $\hat{\boldsymbol{\theta}}^{(t)} = (\hat{\theta}_{t+1}^{(t)}, \ldots, \hat{\theta}_T^{(t)})$, where $\hat{\theta}_i^{(t)} = \hat{d}_i^{(t)}$ represents the demand of quarter/month $i$ estimated on quarter/month $t$. The production and sales for the quarter/month $t$ and the prior $(t - 1)$ quarters/months have already happened and cannot be changed, which are committed variables. There is no penalty function in this scenario. Therefore, the Stage $t$ OP can be formulated as:

$$
\begin{aligned}
\hat{x}^{(t)}, \hat{y}^{(t)} = \arg\max_{x,y} \quad & \sum_{i=1}^{T} p_i y_i - \sum_{i=1}^{T} c_i x_i \\
\text{s.t.} \quad & y_i \leq d_i, \quad \forall i \in \{1, \ldots, t\} \\
& y_i \leq \hat{d}_i^{(t)}, \quad \forall i \in \{t+1, \ldots, T\} \\
& y_i \leq \sum_{j=1}^{i-1} x_j - \sum_{j=1}^{i-1} y_j, \quad \forall i \in \{1, \ldots, T\} \\
& x_i = \hat{x}_i^{(t-1)}, y_i = \hat{y}_i^{(t-1)}, \quad \forall i \in \{1, \ldots, t-1\} \\
& x \geq 0, \quad y \geq 0
\end{aligned}
$$

## D.2 Investment Problem

In the second experiment, we showcase our framework on an MILP. The unknown parameters appear in both the objective and constraints. A person wants to make an investment plan for buying several types of financial products next year to maximize the investment profit, under limited capital. The investment profit contains 3 parts: 1) the interest gained from the products owned, 2) the market prices of the products owned at the end of the year, and 3) profits from selling products minus costs for buying products minus transaction fees. The capital for the whole year is given. However, the market price of each product in each quarter/month is revealed only at the beginning of the quarter/month, and the interest of owning each product in each quarter/month is revealed only at the end of the quarter/month. The person will estimate the market prices and interests based on past experiences, considering features such as the product type, the financial condition and operational capabilities of the company to which the product belongs, and so on.

Suppose there are $T$ quarters/months, and $N$ investment products. The decision variables are: 1) a natural vector $x \in \mathrm{N}^{T \times N}$, where $x_{i,j}$ represents the number of product $j$ on hand at the end of quarter/month $i$, 2) an integer vector $y \in \mathrm{Z}^{(T-1) \times N}$, where $y_{i,j}$ represents the number of product $j$ bought or sold in quarter/month $i$, and 3) a natural vector $z \in \mathrm{N}^{(T-1) \times N}$, where $z_{i,j}$ represents whether the transaction fee is paid for product $j$ in month $i$. Let $p_{i,j}$ denote the interest of product $j$ in month $i$, $w_{i,j}$ denote the market price of product $j$ in month $i$, $C$ denote the capital for the whole year.

We assume that the end of quarter/month $t$, i.e., the beginning of quarter/month $(t + 1)$, is Stage $t$. At Stage 0, i.e., the beginning of quarter/month 1, the person can buy some products without paying a transaction fee. The market price of each product at this time is known, i.e., $w_1 = (w_{1,1}, \ldots, w_{1,N})$ are given. The unknown parameters in this OP are $\boldsymbol{p} \in \mathbb{R}^{T \times N}$ and $\boldsymbol{w} = (w_2, \ldots, w_T) \in \mathbb{R}^{(T-1) \times N}$. At the beginning of each subsequent quarter/month, the person can buy more products or sell the products owned but needs to pay a transaction fee. For simplicity, we assume that the transaction fee for buying/selling product $i$ in quarter/month $j$ is linear in the market price of product $i$ in quarter/month $j$. Here, the linearity factor is independent of the request. That is, if the person buys/sells $k$ number of product $i$ in quarter/month $j$, the person has to spend $k\sigma w_{ij}$, where $\sigma \geq 0$ is a non-negative tunable scalar parameter, and we call it transaction factor.

At Stage 0, i.e., the beginning of quarter/month 1, the person uses the predicted interests $\hat{\boldsymbol{p}}^{(0)}$ and market prices $\hat{\boldsymbol{w}}^{(0)}$ to make the plan. The Stage 0 OP can be formulated as:

$$\hat{x}^{(0)}, \hat{y}^{(0)}, \hat{z}^{(0)} = \underset{x,y,z}{\arg\max} \; obj(\hat{\boldsymbol{p}}^{(0)}, w_1, \hat{\boldsymbol{w}}^{(0)}, x, y, z) \tag{12}$$

$$\text{s.t.} \; \sum_{j=1}^{N} w_{1,j} x_{1,j} \le C, \tag{13}$$

$$\begin{aligned} &\sum_{j=1}^{N} w_{1,j} x_{1,j} \\ &+ \sum_{i=2}^{t}\sum_{j=1}^{N} \sigma \hat{w}_{i,j}^{(0)} z_{i,j} \quad \le C, \quad \forall t \in \{2,\ldots,T\} \\ &+ \sum_{i=2}^{t}\sum_{j=1}^{N} \hat{w}_{i,j}^{(0)} y_{i,j} \end{aligned} \tag{14}$$

$$x_{i,j} = y_{i,j} + x_{(i-1),j}, \quad \forall i \in \{2,\ldots,T\}, j \in \{1,\ldots,N\} \tag{15}$$

$$z_{i,j} \ge y_{i,j}, \quad \forall i \in \{2,\ldots,T\}, j \in \{1,\ldots,N\} \tag{16}$$

$$z_{i,j} \ge -y_{i,j}, \quad \forall i \in \{2,\ldots,T\}, j \in \{1,\ldots,N\} \tag{17}$$

where

$$\begin{aligned} &obj(\hat{\boldsymbol{p}}^{0}, w_1, \hat{\boldsymbol{w}}^{0}, x, y, z) \\ &= \sum_{i=1}^{T} \hat{p}_{i,j}^{(0)} x_{i,j} + \sum_{j=1}^{N} \hat{w}_{T,j}^{(0)} x_{T,j} - \left( \sum_{j=1}^{N} w_{1,j} x_{1,j} + \sum_{i=2}^{T}\sum_{j=1}^{N} \sigma \hat{w}_{i,j}^{(0)} z_{i,j} + \sum_{i=2}^{T}\sum_{j=1}^{N} \hat{w}_{i,j}^{(0)} y_{i,j} \right) \end{aligned}$$

represents the objective, which is to maximize the investment profit; Equations (13) and (14) ensure that the money spent on buying products and transaction fees will not exceed the capital available; Equations (15), (16), and (17) formulate the relationships among three decision variables $x, y,$ and $z$.

At Stage $t$, i.e., the end of quarter/month $t$, the interest of owning each product in quarter/month $t$ as well as the market price of each product revealed. Then, by Stage $t$ $(1 \le t \le T)$, all the true market prices for the prior $t$ quarters/months, as well as the $(t + 1)$ quarter/month, are revealed. Besides, all the true interests for the prior $t$ quarters/months are also revealed. But the market prices for the later $(T - t - 1)$ quarters/months and the interests for the later $(T - t)$ are still uncovered and are estimated as $\hat{\boldsymbol{w}}^{(t)} = (\hat{w}_{t+2}^{(t)}, \ldots \hat{w}_{T}^{(t)})$ and $\hat{\boldsymbol{p}}^{(t)} = (\hat{p}_{t+1}^{(t)}, \ldots \hat{p}_{T}^{(t)})$, where $\hat{w}_i^{(t)}$ and $\hat{p}_i^{(t)}$ represents the market price and the interest of quarter/month $i$ estimated on quarter/month $t$. The investment decisions $x, y, z$ for the prior $t$ quarters/months have already happened and cannot be changed, which are committed variables. There is no penalty function in this scenario.

$$\hat{x}^{(t)}, \hat{y}^{(t)}, \hat{z}^{(t)} = \underset{x,y,z}{\arg\max} \; obj(\boldsymbol{p}[1:t] \oplus \hat{\boldsymbol{p}}^{(t)}, w_1, \boldsymbol{w}[2:t+1] \oplus \hat{\boldsymbol{w}}^{(t)}, x, y, z)$$

$$\text{s.t.} \; \sum_{j=1}^{N} w_{1,j} x_{1,j} \le C,$$

$$\begin{aligned} &\sum_{j=1}^{N} w_{1,j} x_{1,j} \\ &+ \sum_{i=2}^{k}\sum_{j=1}^{N} \sigma w_{i,j} z_{i,j} \quad \le C, \quad \forall k \in \{2,\ldots,t\} \\ &+ \sum_{i=2}^{k}\sum_{j=1}^{N} w_{i,j} y_{i,j} \end{aligned}$$

$$\begin{aligned} &\sum_{j=1}^{N} w_{1,j} x_{1,j} \\ &+ \sum_{i=2}^{t+1}\sum_{j=1}^{N} \alpha w_{i,j} z_{i,j} + \sum_{i=t+2}^{k}\sum_{j=1}^{N} \alpha \hat{w}_{i,j}^{(t)} z_{i,j} \quad \le C, \quad \forall k \in \{t+1,\ldots,T\} \\ &+ \sum_{i=2}^{t+1}\sum_{j=1}^{N} w_{i,j} y_{i,j} + \sum_{i=t+2}^{k}\sum_{j=1}^{N} \hat{w}_{i,j}^{(t)} y_{i,j} \end{aligned}$$

$$x_{i,j} = y_{i,j} + x_{(i-1),j}, \quad \forall i \in \{2,\ldots,T\}, j \in \{1,\ldots,N\}$$

$$z_{i,j} \ge y_{i,j}, \quad \forall i \in \{2,\ldots,T\}, j \in \{1,\ldots,N\}$$

$$z_{i,j} \ge -y_{i,j}, \quad \forall i \in \{2,\ldots,T\}, j \in \{1,\ldots,N\}$$

$$x_{i,j} = \hat{x}_{i,j}^{(t-1)}, \quad \forall i \in \{1,\ldots,t\}, j \in \{1,\ldots,N\}$$

$$y_{i,j} = \hat{y}_{i,j}^{(t-1)}, z_{i,j} = \hat{z}_{i,j}^{(t-1)}, \quad \forall i \in \{2,\ldots,t\}, j \in \{1,\ldots,N\}$$

where

$$obj(\boldsymbol{p}[1:t] \oplus \hat{\boldsymbol{p}}^{(t)}, w_1, \boldsymbol{w}[2:t+1] \oplus \hat{\boldsymbol{w}}^{(t)}, x, y, z)$$

$$= \sum_{i=1}^{t} \sum_{j=1}^{N} p_{i,j} x_{i,j} + \sum_{i=t+1}^{T} \sum_{j=1}^{N} \hat{p}_{i,j}^{(t)} x_{i,j} - \sum_{j=1}^{N} w_{1j} x_{1j} - (\sum_{i=2}^{t+1} \sum_{j=1}^{N} \alpha w_{i,j} z_{i,j} + \sum_{i=t+2}^{T} \sum_{j=1}^{N} \alpha \hat{w}_{i,j}^{(t)} z_{i,j})$$

$$- (\sum_{i=2}^{t+1} \sum_{j=1}^{N} w_{i,j} y_{i,j} + \sum_{i=t+2}^{T} \sum_{j=1}^{N} \hat{w}_{i,j}^{(t)} y_{i,j}) + \sum_{j=1}^{N} \hat{w}_{Tj}^{(t)} x_{Tj}$$

## E   Hyperparameters for the Experiments

The methods of $k$-NN, RF, NN, Baseline, SCD and PCD have hyperparameters, which we tune via cross-validation: for $k$-NN, we try $k \in \{1, 3, 5\}$; for RF, we try different numbers of trees in the forest $\{10, 50, 100\}$; for NN, Baseline, and PCD, we treat the optimizer, learning rate, and epochs as hyperparameters. For Baseline, SCD and PCD, we treat the optimizer, learning rate, the early-cut-off value of log barrier regularization term ($\mu$), and epochs as hyperparameters.

Table 4 show the final hyperparameter choices for the three problems: 1) production and sales problem, 2) investment problem, and 3) nurse rostering problem.

Table 4: Hyperparameters of the experiments on the three problems.

| Model | Hyperaprameters | | |
|---|---|---|---|
| | Production and sales | Investment | Nurse rostering |
| Baseline | optimizer: optim.Adam; learning rate: $1 \times 10^{-7}$; $\mu = 10^{-8}$; epochs=20 | optimizer: optim.Adam; learning rate: $1 \times 10^{-6}$; $\mu = 10^{-8}$; epochs=20 | optimizer: optim.Adam; learning rate: $1 \times 10^{-6}$; $\mu = 10^{-8}$; epochs=20 |
| SCD | optimizer: optim.Adam; learning rate: $1 \times 10^{-7}$; $\mu = 10^{-8}$; epochs=20 | optimizer: optim.Adam; learning rate: $1 \times 10^{-6}$; $\mu = 10^{-8}$; epochs=20 | optimizer: optim.Adam; learning rate: $5 \times 10^{-7}$; $\mu = 10^{-7}$; epochs=20 |
| PCD | optimizer: optim.Adam; learning rate: $1 \times 10^{-7}$; $\mu = 10^{-8}$; epochs=20 | optimizer: optim.Adam; learning rate: $1 \times 10^{-6}$; $\mu = 10^{-8}$; epochs=20 | optimizer: optim.Adam; learning rate: $5 \times 10^{-7}$; $\mu = 10^{-7}$; epochs=20 |
| NN | optimizer: optim.Adam; learning rate: $1 \times 10^{-5}$; epochs=20 | optimizer: optim.Adam; learning rate: $1 \times 10^{-5}$; epochs=20 | optimizer: optim.Adam; learning rate: $1 \times 10^{-3}$; epochs=20 |
| $k$-NN | k=5 | | |
| RF | n estimator=100 | | |

Ridge, $k$-NN, CART and RF are implemented using *scikit-learn* [29]. The neural network is implemented using *PyTorch* [26]. To compute the two stages of optimization at *test time* for our method, and to compute the optimal solution of an (MI)LP under the true parameters, we use the *Gurobi* MILP solver [13].

## F   Detailed Experiment Results

### F.1   Production and Sales Problem

Table 5 reports the mean post-hoc regrets and standard deviations across 30 simulations for all training methods on the production and sales problem. Compared among standard regression models, NN performs well and achieves the best performance in all cases listed in Table 5, while Ridge and RF also have decent performances.

Table 6 shows improvement ratios among the proposed 3 algorithms and BAS. "A vs B" refers to the improvement in the percentage of method A over method B. Take "Baseline vs BAS" as an example, the improvement percentage of the baseline over BAS is $(355.56 - 305.26)/355.56 \times 100\% = 14.15\%$ when $T = 4$ in the low-profit price group. Comparing numbers in "SCD VS BAS", "PCD VS BAS", and "Baseline VS BAS" when stage num = 4 and 12, we can see that the advantages of the proposed methods over BAS are more distinct when the number of stages is larger. Additionally, comparing numbers in "SCD VS Baseline" and "PCD VS Baseline" when stage num = 4 and 12, we also note that the advantages of SCD and PCD over the Baseline are more distinct when the number of stages is larger.

Table 5: Mean post-hoc regrets and standard deviations of all methods for the production and sales problem.

| Price group | Low-profit | | High-profit | |
|---|---|---|---|---|
| Stage num | 4 | 12 | 4 | 12 |
| SCD | **293.78±99.21** | **488.72±127.62** | **505.24±89.55** | **887.38±250.55** |
| PCD | 297.34±107.44 | 495.21±122.42 | 520.76±92.20 | 905.61±255.99 |
| Baseline | 305.26±100.88 | 515.80±137.67 | 526.77±104.99 | 935.03±263.47 |
| NN | 355.56±103.78 | 637.77±199.25 | 561.36±96.49 | 997.44±273.91 |
| Ridge | 390.88±114.89 | 648.60±214.69 | 612.49±109.62 | 1017.41±277.01 |
| knn | 368.20±111.34 | 663.96±208.51 | 591.47±97.87 | 1050.42±296.83 |
| CART | 485.73±152.05 | 873.85±279.68 | 763.88±136.37 | 1345.56±337.05 |
| RF | 375.18±114.23 | 644.63±204.50 | 567.35±94.16 | 1021.51±274.34 |
| TOV | 1615.75±675.77 | 7344.78±2290.04 | 5007.09±976.65 | 21066.00±4159.56 |

Table 6: Improvement ratios among Baseline, SCD, PCD, and standard regression models for the production and sales problem.

| Price group | Stage num | SCD VS BAS | PCD VS BAS | Baseline VS BAS | SCD VS Baseline | PCD VS Baseline | SCD VS PCD |
|---|---|---|---|---|---|---|---|
| Low-profit | 4 | 17.38% | 16.37% | 14.15% | 3.76% | 2.59% | 1.20% |
| | 12 | 23.37% | 22.35% | 19.12% | 5.25% | 3.99% | 1.31% |
| High-profit | 4 | 10.00% | 7.23% | 6.16% | 4.09% | 1.14% | 2.98% |
| | 12 | 11.03% | 9.21% | 6.26% | 5.10% | 3.15% | 2.01% |

Figure 1 shows post-hoc regret comparisons between BAS and the proposed methods (Baseline, SCD, and PCD) for each run. The x-axis refers to the number of each simulation, and the y-axis refers to the ratio of the post-hoc regret achieved by BAS and the proposed methods (Baseline, SCD, and PCD) corresponding to that simulation. To easily read the comparisons, we sorted all simulations by the increasing order of the post-hoc regret ratios of BAS/SCD. The red dashed line where the post-hoc regret ratio is 1.0 represents the boundary line where (Baseline, SCD, or PCD) performs better or worse than BAS. When the point representing the post-hoc regret ratio of BAS/(Baseline, SCD, or PCD) falls above the red dashed line, it means that (Baseline, SCD, or PCD) performs better than BAS. Conversely, when the point falls below the red dashed line, it means BAS performs better than (Baseline, SCD, or PCD). Observing Figure 1, SCD outperforms BAS across all simulations in all 4 scenarios. While not as stable as SCD, PCD and Baseline also outperform BAS in most of the simulations. Compared with Figure 1a, there are more BAS/Baseline points that fall below the red dashed line in Figure 1b, while the number of BAS/SCD points and the number of BAS/PCD points that fall below the red dashed line are similar in Figure 1a and Figure 1b. The same phenomenon can be observed when comparing Figure 1c and Figure 1d, demonstrating the advantage of SCD and PCD over Baseline.

The win rate of a method against other methods can be directly computed from Figure 1 by counting the number of simulations with ratios > 1, and we provide the win rates table as a reference in Table 7. Table 7 indicates that in more than half of the simulations, SCD achieved the best performance. PCD followed closely, achieving the best performance in the remaining 20% to 50% of the simulations.

Table 7: Win rates for the production and sales problem.

| Price group | Stage num | Baseline beats BAS | SCD beats BAS | PCD beats BAS | BAS is best | Baseline is best | SCD is best | PCD is best |
|---|---|---|---|---|---|---|---|---|
| Low-profit | 4 | 93.33% | 96.67% | 86.67% | 0.00% | 3.33% | 50.00% | 46.67% |
| | 12 | 73.33% | 100.00% | 90.00% | 0.00% | 0.00% | 66.67% | 33.33% |
| High-profit | 4 | 66.67% | 96.67% | 73.33% | 0.00% | 23.33% | 50.00% | 26.67% |
| | 12 | 76.67% | 100.00% | 80.00% | 0.00% | 0.00% | 63.33% | 36.67% |

## F.2 Investment Problem

Table 8 and Table 9 report the mean post-hoc regrets and standard deviations across 30 simulations for all training methods on the investment problem. Compared among standard regression models, NN performs well and achieves the best performance in most cases, while Ridge and RF also have decent performances and obtain the smallest mean post-hoc regret in some cases.

Table 10 shows improvement ratios among the proposed 3 algorithms and BAS. Comparing "SCD vs BAS", "PCD vs BAS", and "Baseline vs BAS" performance under the same capital and the same stage number, we observe that the advantages of the proposed methods (SCD, PCD, and Baseline) over the standard regression approaches become more pronounced as the transaction factor increases.

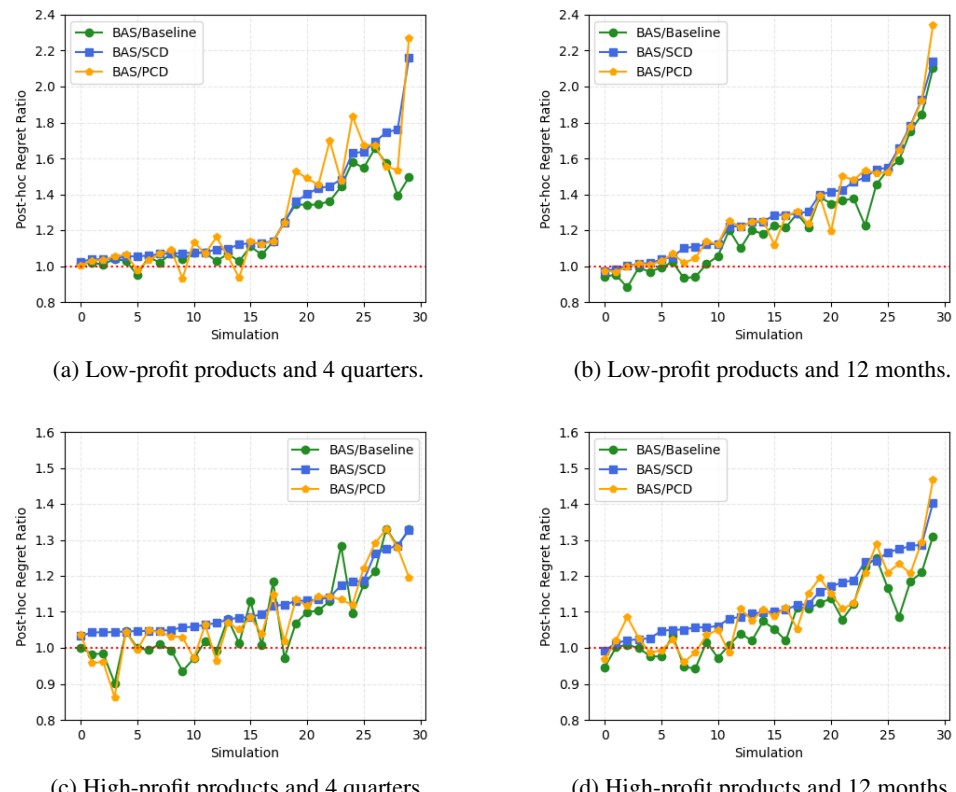

(a) Low-profit products and 4 quarters.

(b) Low-profit products and 12 months.

(c) High-profit products and 4 quarters.

(d) High-profit products and 12 months.

Figure 1: BAS/Baseline, BAS/SCD, and BAS/PCD for the production and sales problem.

Table 8: Mean post-hoc regrets and standard deviations of all methods for the investment problem when capital=25.

| Stage num | 4 | | | 12 | | |
|---|---|---|---|---|---|---|
| Transaction factor | 0.01 | 0.05 | 0.1 | 0.01 | 0.05 | 0.1 |
| SCD | **19.85±3.14** | **14.73±3.59** | **10.56±1.63** | **1513.31±185.03** | **666.96±91.54** | **260.27±34.32** |
| PCD | 20.00±3.24 | 14.90±3.62 | 10.63±1.62 | 1524.69±191.19 | 675.27±95.10 | 263.97±35.09 |
| Baseline | 20.20±3.68 | 15.14±3.62 | 10.77±1.64 | 1540.47±186.90 | 686.84±92.49 | 269.07±34.47 |
| NN | 20.51±3.43 | 15.47±3.67 | 11.23±1.87 | 1563.78±199.67 | 699.30±101.58 | 277.31±32.99 |
| Ridge | 20.88±3.30 | 15.38±3.37 | 11.70±2.00 | 1588.11±200.48 | 703.74±97.62 | 276.51±32.14 |
| $k$-NN | 22.21±3.44 | 16.96±4.18 | 11.56±2.15 | 1643.46±198.96 | 722.73±79.93 | 285.73±41.26 |
| CART | 24.81±4.30 | 19.68±4.58 | 13.42±2.27 | 1845.40±285.85 | 832.02±129.30 | 333.71±51.84 |
| RF | 21.88±3.56 | 16.98±3.74 | 12.07±1.93 | 1563.94±190.01 | 700.31±77.70 | 279.84±34.73 |
| TOV | 64.11±4.91 | 51.53±9.97 | 39.87±2.67 | 2404.22±264.58 | 1147.61±114.54 | 502.05±46.67 |

Besides, comparing "SCD vs Baseline" and "PCD vs Baseline" under the same capital and the same transaction factor but different stage numbers, the advantages of SCD and PCD over Baseline are amplified as the number of stages increases. This trend is consistent with the findings from the experiments on the production and sales problem. One interesting phenomenon is that under the same capital and the same transaction factor, the advantage of the proposed methods over BAS appears to be similar or even less obvious when the number of stages is 12 compared to when it is 4. This observation differs from the pattern seen in the production and sales problem experiments. We hypothesize that this divergence may be attributable to the fundamental differences between the problem settings. While the production and sales problem is a pure LP, the investment problem is an IP with several integrality constraints. The proposed methods relax these integrality constraints and treat the problem as an LP for the purpose of forward and backward propagation. The gaps between the original IP and the relaxed LP may accumulate as the number of stages grows larger, potentially diminishing the advantages of the Predict+Optimize approaches.

Table 9: Mean post-hoc regrets and standard deviations of all methods for the investment problem when capital=50.

| Stage num | 4 | | | 12 | | |
|---|---|---|---|---|---|---|
| Transaction factor | 0.01 | 0.05 | 0.1 | 0.01 | 0.05 | 0.1 |
| SCD | **47.48±6.98** | **34.92±5.57** | **25.50±3.88** | **3846.20±420.94** | **1663.82±208.60** | **646.14±75.52** |
| PCD | 47.67±6.64 | 35.22±5.98 | 25.63±3.93 | 3869.76±420.01 | 1679.17±205.01 | 652.57±74.45 |
| Baseline | 48.24±7.13 | 35.64±6.28 | 25.96±4.64 | 3941.09±437.57 | 1701.51±222.45 | 665.71±76.40 |
| NN | 48.98±7.19 | 36.42±5.70 | 26.62±4.07 | 4046.79±390.52 | 1736.59±232.15 | 680.94±70.76 |
| Ridge | 50.73±6.35 | 37.38±4.66 | 27.62±3.12 | 4019.04±454.78 | 1743.96±217.95 | 682.24±74.91 |
| $k$-NN | 53.49±8.52 | 39.69±7.22 | 28.12±3.86 | 4213.33±434.62 | 1797.32±206.92 | 702.48±94.53 |
| CART | 62.35±11.68 | 47.05±9.14 | 31.81±6.76 | 4723.27±529.86 | 2086.09±325.70 | 835.96±137.08 |
| RF | 52.49±6.73 | 39.07±6.50 | 27.75±3.84 | 3999.70±475.44 | 1748.02±201.68 | 696.46±75.14 |
| TOV | 158.56±11.19 | 126.22±8.86 | 99.83±7.02 | 6166.73±573.51 | 2860.05±288.85 | 1259.99±107.60 |

Table 10: Improvement ratios among Baseline, SCD, PCD, and standard regression models for the investment problem.

| Capital | Stage num | Transaction factor | SCD VS BAS | PCD VS BAS | Baseline VS BAS | SCD VS Baseline | PCD VS Baseline | SCD VS PCD |
|---|---|---|---|---|---|---|---|---|
| 25 | 4 | 0.01 | 3.25% | 2.53% | 1.53% | 1.74% | 1.01% | 0.74% |
| | | 0.05 | 4.23% | 3.15% | 1.57% | 2.70% | 1.61% | 1.12% |
| | | 0.1 | 6.03% | 5.39% | 4.12% | 1.99% | 1.33% | 0.67% |
| | 12 | 0.01 | 3.23% | 2.50% | 1.49% | 1.76% | 1.02% | 0.75% |
| | | 0.05 | 4.07% | 3.17% | 1.78% | 2.90% | 1.68% | 1.23% |
| | | 0.1 | 5.87% | 4.53% | 2.69% | 3.27% | 1.89% | 1.40% |
| 50 | 4 | 0.01 | 3.06% | 2.68% | 1.51% | 1.58% | 1.19% | 0.39% |
| | | 0.05 | 4.12% | 3.29% | 2.14% | 2.02% | 1.18% | 0.85% |
| | | 0.1 | 4.21% | 3.71% | 2.46% | 1.79% | 1.27% | 0.52% |
| | 12 | 0.01 | 3.84% | 3.25% | 1.47% | 2.41% | 1.81% | 0.61% |
| | | 0.05 | 4.19% | 3.31% | 2.02% | 2.22% | 1.31% | 0.91% |
| | | 0.1 | 5.11% | 4.17% | 2.24% | 2.94% | 1.97% | 0.99% |

Table 11 presents the win rate results for Baseline, SCD, PCD, and BAS. We observe that SCD outperforms BAS across most of the simulations, with win rates exceeding 80% in most scenarios. PCD also shows strong performance, particularly at higher transaction factors. Compared with BAS, PCD achieves win rates above 80% in many scenarios, closely following SCD.

Table 11: Win rates for the investment problem.

| Capital | Stage num | Transaction factor | Baseline beats BAS | SCD beats BAS | PCD beats BAS | BAS is best | Baseline is best | SCD is best | PCD is best |
|---|---|---|---|---|---|---|---|---|---|
| 25 | 4 | 0.01 | 53.33% | 86.67% | 73.33% | 3.33% | 30.00% | 46.67% | 20.00% |
| | | 0.05 | 66.67% | 90.00% | 86.67% | 0.00% | 33.33% | 43.33% | 23.33% |
| | | 0.1 | 70.00% | 93.33% | 90.00% | 0.00% | 33.33% | 46.67% | 20.00% |
| | 12 | 0.01 | 66.67% | 93.33% | 83.33% | 0.00% | 3.33% | 73.33% | 23.33% |
| | | 0.05 | 80.00% | 96.67% | 93.33% | 0.00% | 6.67% | 83.33% | 10.00% |
| | | 0.1 | 83.33% | 100.00% | 96.67% | 0.00% | 0.00% | 86.67% | 13.33% |
| 50 | 4 | 0.01 | 60.00% | 80.00% | 66.67% | 3.33% | 23.33% | 43.33% | 30.00% |
| | | 0.05 | 66.67% | 93.33% | 83.33% | 3.33% | 36.67% | 40.00% | 20.00% |
| | | 0.1 | 70.00% | 96.67% | 90.00% | 0.00% | 26.67% | 43.33% | 30.00% |
| | 12 | 0.01 | 70.00% | 83.33% | 83.33% | 3.33% | 0.00% | 80.00% | 16.67% |
| | | 0.05 | 73.33% | 90.00% | 86.67% | 3.33% | 0.00% | 86.67% | 10.00% |
| | | 0.1 | 76.67% | 100.00% | 90.00% | 0.00% | 0.00% | 93.33% | 6.67% |

## F.3 Nurse Rostering Problem

Table 12 reports the mean post-hoc regrets and standard deviations across 30 simulations for all training methods on the nurse rostering problem. Compared among standard regression models, NN always performs well and achieves the best performance, while Ridge and RF also have decent performances.

Table 13 shows improvement ratios among the proposed 3 algorithms and BAS. Comparing "SCD vs BAS", "PCD vs BAS", and "Baseline vs BAS" performance with different extra nurse payments, we observe that the advantages of the proposed methods (SCD, PCD, and Baseline) over the standard regression approaches become more pronounced as the extra nurse payment increases.

Figure 2 shows post-hoc regret comparisons between BAS and the proposed methods (Baseline, SCD, and PCD) for each run. To easily read the comparisons, we again sorted all simulations by the increasing order of the post-hoc regret ratios of BAS/SCD. Observing Figure 2, the proposed

Table 12: Mean post-hoc regrets and standard deviations of all methods for the nurse rostering problem.

| Extra nurse payment | 15 | 20 | 25 | 30 |
|---|---|---|---|---|
| SCD | **607.66±142.19** | **789.65±200.22** | **1038.29±255.42** | **1207.50±319.25** |
| PCD | 622.05±153.64 | 805.11±224.99 | 1048.08±281.32 | 1240.48±332.39 |
| Baseline | 629.35±153.67 | 817.60±219.47 | 1083.45±259.68 | 1290.10±371.08 |
| NN | 662.34±169.17 | 863.02±214.50 | 1144.63±305.00 | 1369.81±373.20 |
| Ridge | 663.57±141.49 | 887.63±206.36 | 1146.56±297.33 | 1371.20±320.37 |
| $k$-NN | 758.49±135.75 | 1033.88±197.22 | 1309.92±255.86 | 1562.98±298.14 |
| CART | 965.16±207.67 | 1303.68±280.11 | 1645.59±350.32 | 1957.13±433.46 |
| RF | 680.47±148.20 | 870.50±221.81 | 1145.32±293.18 | 1378.68±333.73 |
| TOV | 10611.64±1574.11 | 10732.32±1504.12 | 10893.54±1485.37 | 11110.73±1344.15 |

Table 13: Improvement ratios among Baseline, SCD, PCD, and standard regression models for the nurse rostering problem.

| Extra nurse payment | SCD VS BAS | PCD VS BAS | Baseline VS BAS | SCD VS Baseline | PCD VS Baseline | SCD VS PCD |
|---|---|---|---|---|---|---|
| 15 | 8.26% | 6.08% | 4.98% | 3.45% | 1.16% | 2.31% |
| 20 | 8.50% | 6.71% | 5.26% | 3.42% | 1.53% | 1.92% |
| 25 | 9.29% | 8.44% | 5.35% | 4.17% | 3.26% | 0.93% |
| 30 | 11.85% | 9.44% | 5.82% | 6.40% | 3.85% | 2.66% |

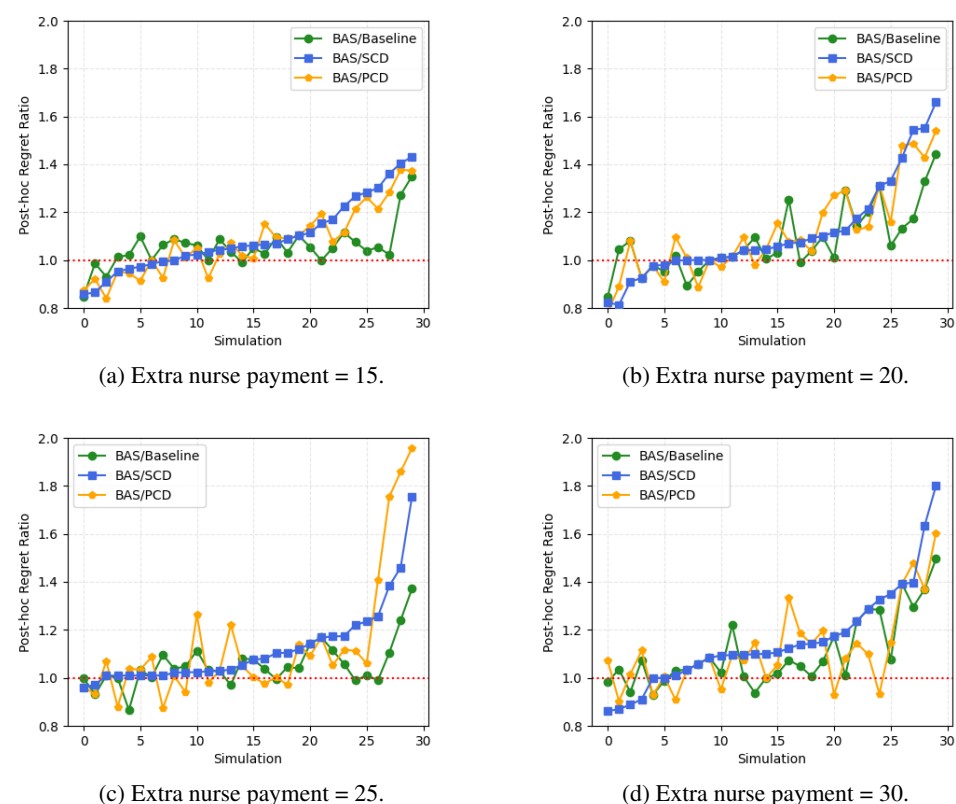

(a) Extra nurse payment = 15.

(b) Extra nurse payment = 20.

(c) Extra nurse payment = 25.

(d) Extra nurse payment = 30.

Figure 2: BAS/Baseline, BAS/SCD, and BAS/PCD for the nurse rostering problem.

methods outperform BAS in most of the simulations. Since the nurse rostering problem is an IP with several integrality constraints, and the proposed methods just relax these constraints and treat the problem as an LP for the purpose of forward and backward propagation. We hypothesize that the gap between the original IP and the relaxed LP may diminish the advantages of the proposed methods, and thus, BAS sometimes performs slightly better than the proposed methods.

Table 14 presents the win rate results for Baseline, SCD, PCD, and BAS. Same as in the first two benchmarks, SCD demonstrates high win rates against BAS across all scenarios. SCD achieves win rates of 86.67% or higher in three out of four cases, peaking at 96.67% when the extra payment is 25. PCD also performs well, with win rates ranging from 70% to 83.33% against BAS. While not as consistently high as SCD, PCD's performance remains competitive, indicating its viability as an alternative approach.

Table 14: Win rates for the nurse rostering problem.

| Extra nurse payment | Baseline beats BAS | SCD beats BAS | PCD beats BAS | BAS is best | Baseline is best | SCD is best | PCD is best |
|---|---|---|---|---|---|---|---|
| 15 | 70.00% | 70.00% | 70.00% | 10.00% | 26.67% | 40.00% | 23.33% |
| 20 | 73.33% | 86.67% | 80.00% | 6.67% | 10.00% | 50.00% | 33.33% |
| 25 | 73.33% | 96.67% | 83.33% | 3.33% | 16.67% | 43.33% | 36.67% |
| 30 | 73.33% | 86.67% | 76.67% | 3.33% | 6.67% | 60.00% | 30.00% |

# G  Runtimes for the Experiments

Table 15: Average training time (in seconds) for the three benchmarks (in seconds).

| Training time (s) | Production and sales | | | | Investment | | | | Nurse rostering |
|---|---|---|---|---|---|---|---|---|---|
| | Stage num = 4 | | Stage num = 12 | | Stage num = 4 | | Stage num = 12 | | Stage num = 7 |
| | Low-profit | High-profit | Low-profit | High-profit | Capital = 25 | Capital = 50 | Capital = 25 | Capital = 50 | \ |
| SCD | 828.79±216.69 | 700.63±244.37 | 3287.99±809.72 | 2552.73±991.69 | 402.37±58.00 | 535.18±88.45 | 7734.01±1198.41 | 11216.01±1994.75 | 14949.59±3281.24 |
| PCD | 293.41±96.27 | 236.80±81.07 | 483.28±95.81 | 470.76±124.97 | 157.25±41.65 | 194.40±57.51 | 2639.72±648.22 | 4509.83±767.45 | 6801.54±1175.01 |
| Baseline | 157.72±50.85 | 100.09±32.50 | 195.42±35.03 | 169.58±45.62 | 56.04±14.73 | 61.49±18.30 | 669.01±261.78 | 797.61±282.70 | 2618.63±524.37 |
| NN | 70.58±24.78 | | 97.60±46.17 | | 49.24±18.24 | | 70.81±29.81 | | 61.41±5.28 |
| Ridge | 1.71±0.29 | | 2.88±0.39 | | 5.60±1.28 | | 17.45±7.96 | | |
| $k$-NN | 0.98±0.96 | | 1.03±0.24 | | 1.92±0.62 | | 11.35±0.99 | | |
| CART | 0.77±0.19 | | 2.46±0.39 | | 5.79±1.45 | | 27.30±2.19 | | ≤ 1 |
| RF | 24.82±1.13 | | 91.93±1.80 | | 358.19±4.26 | | 1150.64±484.87 | | |

In this paper, all models are trained with Intel(R) Xeon(R) CPU E5-2630 v2 @ 2.60GHz processors on Google Colab. Since the testing time of different approaches is quite similar and close to being negligible, here, we only show the training time of each prediction model and do not include the testing time. At training time, the proposed Baseline, SCD, and PCD methods need to solve the LP. Training for the usual NN does not involve the LP at all, and so training is much faster (but gives much worse results).

There are two stopping criteria for SCD and PCD. We set the maximum iteration number of SCD and PCD as 5. Besides, if the difference between the post-hoc regret of two consecutive iterations is less than 1, we consider that the algorithm has converged. This is the second stopping criterion.

Table 15 shows the average training time across 30 simulations for the three problems. For the investment problem, since the training times under different transaction fees are similar when the capital and the number of stages are the same, we do not report them one by one but report only the average. For the nurse rostering problem, since the training times under different extra nurse payments are similar when the numbers of stages are the same, we also do not report them one by one but report only the average.

Since the proposed 3 methods require solving multiple LPs when training, their training times are usually longer than standard methods. But since the production and sales problem is an LP, the solving time of which is not too long, the training time of Baseline is around double of NN. Table 15 also shows that SCD and PCD usually converge in 4 iterations in the production and sales problem.

In the investment problem, the training times of Baseline are better than that of RF. The solving time of the OP, i.e., the difficulty of solving the OP, can largely affect the training times of the proposed methods. When the number of stages grows larger, the investment problem is more difficult to solve. Therefore, the training times of Baseline when there are 4 stages are quite comparable with that of NN, but the training times of Baseline when there are 12 stages are much larger than that of NN. In addition, when the OP becomes more complex, the number of iterations required for SCD and PCD convergence also increases. SCD and PCD convergence usually in 2-3 iterations when there are 4 stages, and usually in 3-4 iterations when there are 12 stages.

In the NRP, since the solving time of 1 NRP is large, the training times of the proposed methods are larger than standard regression methods, which indicates that one future research direction is the speed-vs-prediction accuracy tradeoffs on Multi-Stage Predict+Optimize.

# H  Other End-to-End Training Approaches on MILPs

As mentioned in Section 3 and Section 4, other types of training algorithms can be used within the proposed framework. Considering the performance in both post-hoc regret and training time, we proposed three training algorithms in Section 4. However, there are additional possibilities, and we will discuss two of them in this section.

## H.1  Revealed Parameters as a Feature for Later Stage Predictions

As mentioned in Section 3, the proposed framework allows the training algorithm to choose whether to incorporate the revealed true parameters as additional features as input. In the current implementation of the proposed SCD and PCD methods, we did not include these revealed parameters because, in our preliminary experiments, including them does not really improve prediction quality while just increasing training time.

Table 16 reports the results from the experiments on the production and sales problem by incorporating the revealed parameters from stage $t - 1$ as inputs to the stage $t$ networks. As Table 16 shows, the performance using this expanded input did not improve over the results of the proposed three approaches.

Table 16: Mean post-hoc regrets and standard deviations of training SCD with revealed parameters, training PCD with revealed parameters, SCD, PCD, and Baseline for the production and sales problem..

| Price group | Low-profit | | High-profit | |
|---|---|---|---|---|
| Stage num | 4 | 12 | 4 | 12 |
| SCD with revealed | 294.44±81.72 | 488.93±116.51 | 507.30±63.74 | 890.48±153.28 |
| PCD with revealed | 299.73±82.91 | 498.32±123.25 | 521.82±74.86 | 911.04±185.57 |
| SCD | 293.78±99.21 | 488.72±127.62 | 505.24±89.55 | 887.38±250.55 |
| PCD | 297.34±107.44 | 495.21±122.42 | 520.76±92.20 | 905.61±255.99 |
| Baseline | 305.26±100.88 | 515.80±137.67 | 526.77±104.99 | 935.03±263.47 |

The reason that including revealed parameters does not lead to better predictive performance is due to a combination of the nature of the data and the neural network architecture. Consider one extreme: the true parameter vectors in every single stage are always equal. A neural network model (or other reasonable models) that takes in the prior-stage true parameters should be able to learn to pick up on that and use the information. Consider the other extreme: if the parameter vectors are completely independent across stages, then no model will be able to use prior-stage true parameter information, since no actual information exists. Reality (and our particular dataset) is somewhere in between: there can be some correlation between stages, but, it really depends on whether such information is extractable by the neural network architecture (or whatever other prediction model one wants to use within our multi-stage framework).

For general applications, the decision whether to use the prior-stage true parameters is essentially another hyperparameter, to be tuned for that particular application using the available training data (though one can be safe and always include it, at the expense of training time).

## H.2  Training Different-Stage Neural Networks Simultaneously (or, Necessity of Coordinate Descent Approach)

We propose two coordinate descent based methods in Section 4, but it is also possible to train all networks simultaneously, without the use of coordinate descent. In this section, we compare the proposed two methods against a simultaneous training method, to better motivate the necessity of coordinate descent.

A reasonable way (and the only way we can think of) to train all networks simultaneously is replace prior and future stage predictions in SCD and PCD by the ground truth parameters. However, intuitively, this is a worse approach than the proposed methods, given the interdependency of the predictors: the performance of a predictor depends on the predictions and choices made in past and future stages.

Table 17: Mean post-hoc regrets and standard deviations of a simultaneous training method, SCD, PCD, and Baseline for the production and sales problem..

| Price group | Low-profit | | High-profit | |
|---|---|---|---|---|
| Stage num | 4 | 12 | 4 | 12 |
| SCD | 293.78±99.21 | 488.72±127.62 | 505.24±89.55 | 887.38±250.55 |
| PCD | 297.34±107.44 | 495.21±122.42 | 520.76±92.20 | 905.61±255.99 |
| Simultaneously training | 300.28±103.85 | 509.07±129.93 | 523.38±88.52 | 925.35±223.17 |
| Baseline | 305.26±100.88 | 515.80±137.67 | 526.77±104.99 | 935.03±263.47 |

We have explored this alternative training method and here are the preliminary results on the production and sales problem in Table 17. We observe that the solution quality achieved by the simultaneous training method was worse than the SCD and PCD methods.

The coordinate descent strategy in Section 4 was needed to capture the complex interactions between the networks in different stages, and crucial for the strong performance.

