# OpenReview forum: "Multi-Stage Predict+Optimize for (Mixed Integer) Linear Programs"
_NeurIPS.cc/2024/Conference — NeurIPS 2024 poster_

### Official Review · Reviewer_fVWn · 2024-07-10

**Soundness:** 4
**Presentation:** 3
**Contribution:** 3
**Rating:** 6
**Confidence:** 4

**Summary:**

This paper addresses optimization problems with parameter values unknown at solving time. Specifically, the work takes a supervised learning approach to predicting these parameters, which are revealed over multiple “stages.” The authors extend the Predict+Optimize framework, which uses an optimization-inspired loss function rather than typical mean squared error. This paper addresses multi-stage MILP problems, where unknown parameters of the optimization problem are revealed over several sequential stages. Three algorithms are proposed for training the prediction model, including the joint training of individual neural network predictors for each stage of the problem. Finally, experiments are conducted on three benchmark problems showing the advantage of the new framework over classical training methods.

**Strengths:**

- The multi-stage framework is well motivated as an extension of Two-Stage Predict+Optimize, and the paper is well-written. The notation and presentation of the training methods is easy to follow.
- The authors perform extensive computational experiments, finding that the proposed coordinate descent training algorithms perform well in the benchmarks presented in the paper.
- The trade-off between prediction accuracy and training times for the training algorithms is clearly shown.

**Weaknesses:**

- While this work is very interesting, the fit to this venue is questionable. Firstly, the value of this work to the ML community is unclear; it seems the main impact is in operations research. Secondly, many crucial aspects are moved to the appendices, e.g., all of the optimization problems and most of the computational results. For these reasons, perhaps a leading operations research journal would provide more space and be more appropriate.
- While the integration of machine learning and optimization is an increasingly popular field, this paper does not reference many works. Adding references around using ML to help mixed-integer programming and especially stochastic programming, e.g., in Appendix A.3 or elsewhere, would better help readers place this work in the context.
- The termination criteria for the algorithms in Algorithm 1 are omitted. They are also not discussed in Section 4.2-4.3, i.e., definition of convergence.
- Training the neural networks for these problems requires differentiating through a MILP, which is approximated by differentiating its convex relaxation. The authors do not discuss the quality of this approximation, such as if the strength of the convex relaxation impacts the performance. Often, multiple MILP formulations are possible, with different convex relaxations.

**Questions:**

- Line 109: how common is the assumption that all the variables can be changed in Stage 1 (i.e., when does this occur)? This is what enables the guarantee of feasibility, unlike the hard commitments introduced later in the paper.
- Line 157: likewise, what are the implications of the assumption that future decisions are always feasible given any choice of hard commitments? This should be discussed as a limitation.
- Line 163: please check this. The two-stage framework in Section 2 includes no hard commitments.

**Limitations:**

Yes

---

> ### Author Rebuttal · Authors · 2024-08-07
>
> Thank you for your positive review of our work. We respond to your questions and comments below.
>
> **Venue fit**: While Predict+Optimize does have some operations research flavor, it is fundamentally a machine learning problem nonetheless. We also note that the NeurIPS+ICML community has shown interest in this line of work. This is evidenced by prior works that are specifically on the topic of Predict+Optimize that have been accepted in past ICML or NeurIPS ([1,2] in ICML 2022, and [3] in NeurIPS 2023). Additionally, papers that provide tools that we use, for example works studying differentiation through optimization layers, are widely accepted as part of the community. We hope that, by presenting a new use case, our work can inspire further development of such tools. In summary, we believe there is value in disseminating the current work at NeurIPS, and so we made the decision to submit here.
>
> **Additional literature**: Thanks for the suggestion, we are happy to add these references in Appendix A.3 to situate our work in this broader context.
>
> Specifically, we will include references to prior work on integrating machine learning with stochastic programming, such as [4] proposes an end-to-end learning framework that directly aligns the training of probabilistic machine learning models with the ultimate task-based objective in the context of stochastic programming. [5] uses a neural network to approximate the expected value function in two-stage stochastic programming problems, enabling more efficient solution approaches compared to traditional methods. [6] proposes a neural network-based stagewise decomposition algorithm that can effectively approximate value functions for large-scale multistage stochastic programming problems. [7] develops a principled end-to-end learning framework for training neural network decision maps that can effectively solve stochastic optimization problems, including empirical risk minimization and distributionally robust optimization formulations.
>
> Furthermore, we will ensure to highlight relevant work on using ML to help mixed-integer programming (MIP): ML algorithms for exact solving MIP by branch-and-cut based algorithms [8,9,10], ML algorithms for exact solving MIP by decomposition-based algorithms [11,12], ML algorithms for approximate solving MIP by large neighborhood search based algorithm [13,14], and so on.
>
>
> **Termination criterion**: Thanks for pointing out our accidental omission. Currently, we use a termination criterion which is to threshold by the difference in training set post-hoc regrets between two (outermost) iterations of the training coordinate-descent. The threshold we used is 0.1, but that is just another hyperparameter to be tuned per each application. We will add this info back into the paper.
>
> **Convex relaxation of integrality constraints**: This is an excellent point that deserves further investigation, and we agree that it can be a fruitful direction for future work. As we pointed out in the Future Works section in lines 355-362, the experimental results suggest that our methods exhibit rather different behaviors depending on whether the underlying optimization problem is a linear program or a mixed integer program. We were wondering in a similar direction, as to how the relaxation might affect the prediction performance. Exploring the impact of the choice of convex relaxation on the overall performance of our framework is an important direction for future research.
>
> **Line 109, and Line 163**: We would like to clarify that this is an assumption made by Hu et al. in their Two-Stage framework, and not an assumption we make in our new framework. Regardless, from our reading of Hu et al.'s work, they discussed how their framework can nonetheless model decision variables that cannot be changed in Stage 1 (i.e. hard commitments made in Stage 0), via choosing a penalty function that yields infinite penalty when changing hard commitments already made in Stage 0. They discuss the modelling approach/choice in Appendix A.1 in their paper.
>
> **Line 157, always-feasible assumption**: Please refer to our response to Reviewer Bz3V.
>
> [1] Jeong, Jihwan, et al. "An exact symbolic reduction of linear smart predict+ optimize to mixed integer linear programming." ICML 2022.
>
> [2] Mandi, Jayanta, et al. "Decision-focused learning: Through the lens of learning to rank." ICML 2022.
>
> [3] Hu, Xinyi, Jasper Lee, and Jimmy Lee. "Two-Stage Predict+ Optimize for MILPs with Unknown Parameters in Constraints." NeurIPS 2023.
>
> [4] Donti, Priya, Brandon Amos, and J. Zico Kolter. "Task-based end-to-end model learning in stochastic optimization." NeurIPS 2017.
>
> [5] Patel, Rahul Mihir, et al. "Neur2SP: Neural two-stage stochastic programming." NeurIPS 2022.
>
> [6] Bae, Hyunglip, et al. "Deep value function networks for large-scale multistage stochastic programs." AISTATS 2023.
>
> [7] Rychener, Yves, Daniel Kuhn, and Tobias Sutter. "End-to-end learning for stochastic optimization: A bayesian perspective." ICML 2023.
>
> [8] Balcan, Maria-Florina, et al. "Learning to branch." ICML 2018.
>
> [9] Gasse, Maxime, et al. "Exact combinatorial optimization with graph convolutional neural networks." NeurIPS 2019.
>
> [10] Zarpellon, Giulia, et al. "Parameterizing branch-and-bound search trees to learn branching policies." AAAI 2021.
>
> [11] Lange, Jan-Hendrik, and Paul Swoboda. "Efficient message passing for 0–1 ILPs with binary decision diagrams." ICML 2021.
>
> [12] Lozano, Leonardo, David Bergman, and J. Cole Smith. "On the consistent path problem." Operations Research, 68.6, 2020.
>
> [13] Song, Jialin, Yisong Yue, and Bistra Dilkina. "A general large neighborhood search framework for solving integer linear programs." NeurIPS 2020.
>
> [14] Liu, Defeng, Matteo Fischetti, and Andrea Lodi. "Learning to search in local branching." AAAI 2022.

---

> ### Comment · Reviewer_fVWn · 2024-08-11
> **Response to rebuttal**
>
> I have read the authors responses and thank the authors for their answers. I maintain my score and overall positive impression of this work. Note that it would indeed make this work more comprehensive to investigate the effect of the tightness of the convex relaxation, as this can directly affect the accuracy of the trained surrogate models.

---

> > ### Author Response · Authors · 2024-08-11
> >
> > Thank you for appreciating that our work is very interesting and well-motivated. We definitely agree with you as well that the effect of the tightness of convex relaxation is important future work.

---

### Official Review · Reviewer_Bz3V · 2024-07-12

**Soundness:** 3
**Presentation:** 3
**Contribution:** 2
**Rating:** 6
**Confidence:** 2

**Summary:**

The paper proposes a Multi-Stage Predict+Optimize framework that addresses optimization problems where parameters are revealed in sequential stages. It introduces three neural network training algorithms tailored for mixed integer linear programs (MILPs) under this framework. The methodologies are empirically evaluated using three benchmarks, demonstrating enhanced predictive performance over classic methods.

**Strengths:**

1. **Innovative Framework**: The proposed multi-stage framework is a significant extension of the existing two-stage predict+optimize models, effectively addressing scenarios where parameters are revealed progressively.
2. **Robust Empirical Evaluation**: The paper provides a comprehensive experimental section that not only demonstrates the superiority of the proposed methods over traditional approaches but also discusses the trade-offs between them.

**Weaknesses:**

1. **Assumptions on Constraints**: The paper assumes that stage-wise optimizations are always feasible regardless of previous stages' outcomes and current parameter estimates. This assumption may not hold in practical scenarios, potentially limiting the framework's applicability.

2. **Limited Benchmarks**: The choice of benchmarks does not include complex problems like the Alloy Production Problem or the 0-1 knapsack[1], which could benefit from a multi-stage approach. This omission raises questions about the framework's applicability to a broader range of problems.

[1] X. Hu, J. C. H. Lee, and J. H. M. Lee. Two-Stage Predict+Optimize for mixed integer linear programs with unknown parameters in constraints. In Advances in Neural Information Processing Systems, 2023.

**Questions:**

1. **Parameter Inclusion in Predictions**: Why aren't revealed parameters included in subsequent predictions? Would incorporating these parameters enhance prediction accuracy? I hope the author will share more of your thoughts on this with us.

2. **Clarification on Terminology**: What does TOV stand for in the last columns of Tables 1 and 2? I could not find the full form of TOV in the main text.

**Limitations:**

1. **Applicability Concerns**: Despite listing applications like nurse scheduling, the practical applicability across diverse real-world scenarios remains uncertain. More concrete examples and a broader range of applications would strengthen the claims.

2. **Innovativeness**: While the paper extends the two-stage predict+optimize framework to multi-stage, the core ideas closely resemble the existing models, which might diminish the perceived novelty.

**Additional Note on Writing Corrections**:
- Correct the notation for $\hat{x}_0$ to $\hat{x}^{(0)}$ where applicable in line 101.
- Ensure consistency in table font sizes, especially noting that Table 3’s font size is noticeably smaller than that in Tables 1 and 2.

This is an interesting field, and if the authors can provide sufficiently detailed responses to the questions raised, I would be willing to consider revising my score.

---

> ### Author Rebuttal · Authors · 2024-08-07
>
> Thank you for your review and your appreciation of the significance of our work. We address your concerns in this individual response.
>
> **Assumption on constraints**: As the reviewer pointed out, we make an assumption that the optimization problems are always feasible, which seems like a strong assumption on the surface. However, we argue that this is in fact natural, and essentially a necessary assumption in practice. In real-life applications, if we encounter an unsatisfiable scenario, this means something catastrophic is going to happen. Before actually using the application, the domain expert should always have designed the underlying real-world system to have recourse actions to mitigate bad prior actions (at cost/penalty) and to prevent catastrophe, and furthermore model such recourse actions in the (multi-stage) optimization problem. Any system and corresponding formulation of multi-stage optimization problem without such recourse should never be used/executed in the first place. It is thus a reasonable assumption and a practical responsibility we ask of the practitioner, that recourse actions are always designed into the underlying system and modelled, so that our feasibility assumption is satisfied.
>
> **Benchmarks**: We want to point out that the benchmarks we consider are more complex than those in [1].
>
> The production and sales problem we consider is not only a multi-stage extension of the alloy production problem in [1], but is more complex. The alloy production problem in [1] is a pure packing linear program, while our production and sales problem is a general (non-packing) linear program.
>
> The investment problem we consider is also a more complex variant of the multi-stage 0-1 knapsack problem. In the 0-1 knapsack problem in [1], the objective function only considers maximizing profits. In our investment problem, the objective function considers profits from buying and selling, as well as interest gains. Additionally, the constraints in our investment problem are also more intricate.
>
> **Parameter Inclusion in Predictions**: Please see overall response.
>
> **TOV**: TOV stands for true optimal value (optimal objective value under true parameters in hindsight). Thanks for pointing out that we forgot to define this.
>
> **Applicability Concerns**: The investment problem and the production and sales problem benchmarks are also applicable in real-life. As explained above, these benchmarks are more complex and realistic than the benchmarks considered by Hu et al. in their Two-Stage paper.
>
> **Innovativeness**: Please see our responses to Reviewers GGkj and aupq.

---

> > ### Comment · Reviewer_Bz3V · 2024-08-10
> > **Response to Rebuttal**
> >
> > Firstly, I would like to thank the authors for their detailed responses to my review comments. I appreciate the effort made to address the concerns raised.
> >
> > Regarding the decision not to include the revealed parameters from stage $t−1$ as inputs for predictions at stage $t$, the authors mentioned that preliminary experiments showed that including these parameters did not significantly improve prediction quality but did increase training time. This result seems counterintuitive, as incorporating more accurate revealed information would typically be expected to guide and correct current predictions more effectively. Could the authors provide further analysis on why including these revealed parameters did not lead to better predictive performance? Is it possible that this outcome is due to specific experimental settings, the complexity of the model architecture, or the nature of the data itself? A deeper exploration of this phenomenon could provide valuable insights and help in further understand the model's performance.
> >
> > Thank you again for your thoughtful responses. I look forward to your further clarifications.

---

> > > ### Author Response · Authors · 2024-08-10
> > >
> > > We thank the reviewer for appreciating our rebuttal explanations, and for following up.
> > >
> > > Re: parameter inclusion in next stage prediction. Thank you for pushing us further on this, this is worth adding discussion in the revised paper. For your question, the outcome necessarily has to be due to a combination of the nature of the data and the neural network architecture.
> > >
> > > Consider one extreme: the true parameter vectors in every single stage are always equal. A neural network model (or other reasonable models) that takes in the prior-stage true parameters should be able to learn to pick up on that and use the information.
> > >
> > > Consider the other extreme: if the parameter vectors are completely independent across stages, then no model will be able to use prior-stage true parameter information, since no actual information exists.
> > >
> > > Reality is probably somewhere in between: there can be some correlation between stages, but, it really depends on whether such information is extractable by the neural network architecture (or whatever other prediction model one wants to use within our multi-stage framework). So the decision whether to use the prior-stage true parameters is essentially another hyperparameter, to be tuned per application using the available training data (though one can be safe and always include it, at the expense of training time).
> > >
> > > In order to address this concern, in the revised manuscript we will:
> > > - Explicitly include in our framework the possibility of using the prior-stage parameters as input to the models.
> > > - Add the above discussion, which can help guide practitioner intuition.
> > >
> > > Please let us know if you have any further questions! Thank you again for engaging with us.

---

> > > > ### Comment · Reviewer_Bz3V · 2024-08-13
> > > > **Response to Rebuttal**
> > > >
> > > > Thank you for your excellent response. Your explanation has addressed my concerns and clarified the rationale behind the decision regarding parameter inclusion. Although I am not deeply familiar with this specific area, I appreciate the thoroughness of your reply and the contributions you have made. Based on your clarifications, I will be raising my score.
> > > >
> > > > Thank you again for your hard work and dedication to this research.

---

> > > > > ### Author Response · Authors · 2024-08-14
> > > > >
> > > > > We thank the reviewer again for the useful discussion, and we're glad that you appreciate our paper and our careful and thorough rebuttals/responses.

---

### Official Review · Reviewer_aupq · 2024-07-12

**Soundness:** 2
**Presentation:** 3
**Contribution:** 2
**Rating:** 3
**Confidence:** 3

**Summary:**

The authors present an approach for learning hidden parameters for multi-stage optimization problems where parameters are gradually revealed at each stage. In this setting, latent parameters are predicted, then soft committed decisions are made based on those predictions, that stage’s parameters are revealed, and the practitioner can modify the soft committed solution to a hard committed solution for a penalty. Once that hard committed solution is fixed, the next stage begins with new predictions. After all stages are done, the post hoc regret is determined by the difference between the quality of decisions made and the optimal hindsight sequence of decisions, minus incurred solution modification penalties.

The goal is to train models that predict the next time step’s parameters, such that the overall regret is minimized. To solve this problem, the authors propose generalizing previous work on end-to-end training for the same post hoc regret developed for two stage problems. Here, the authors consider unrolling the multiple stages and propose three methods of training: baseline, coordinate descent, and parallel coordinate descent. The baseline approach considers the same model to be predicting at each stage. The other two approaches consider training independent networks for the different stages. Coordinate descent trains each network individually by fixing the other network weights and caching intermediate solutions if they don’t need to be recomputed. Parallel coordinate descent trains the different networks in parallel disregarding that the gradients for a given network may not be representative of the gradients obtained given the state of the other models.

The authors run experiments on three synthetic domains motivated by real world problems and show that their approaches outperform optimization-agnostic learning methods.

**Strengths:**

1 The authors tackle an interesting problem in multi-stage contextual optimization, generalizing previous work on two stages to multiple stages

2 The authors present three approaches for training the predictive model, varying what model is considered and which network weights are frozen, with various tradeoffs in performance and training time

3 the authors evaluate performance on three domains motivated by real world problems with nontrivial optimization formulations showing improved performance over standard optimization-agnostic approaches

**Weaknesses:**

1 the problem settings while well motivated are synthetic and seem to be pulling data from sources that are quite unrelated to the problem domain. For instance, the ICON challenge used for nurse scheduling represents energy data. Additionally, it is unclear what the knapsack data is supposed to represent and why it would be relevant for the oil shipment or investment settings. For the investment settings in particular, previous work [1,2,3,4] has used real world public stock data to evaluate performance. Given that the contribution is mostly empirical, it would be helpful to demonstrate performance on real-world settings with relevant data.

2 The approaches themselves are somewhat incremental considering previous work. Mainly, the approaches consider unrolling the two-stage approach considered in previous work.

4 It is unclear how substantial the performance improvements are given that the variability is quite large and induce overlapping confidence intervals. It might be helpful to give win rate results to determine how often a given approach is the top-performing approach.

5 It is unclear why a coordinate descent approach is needed. Would it be possible to train all networks for all timesteps simultaneously? It would help to better motivate the necessity of coordinate descent by comparing against a simultaneous training approach.

Minor comments

Formal framework definition - it would be good to keep the revealed (and predicted) indices consistent. The off-by-one indices appear to be frequently mixed up. For instance, line 142, true parameters theta1,…, theta t-1 are revealed but then line 144 - theta t is revealed. Additionally, soft solutions \hat{x} seem to start at 0 but are indexed from 1 to t-1 in line 152. It is fairly clear from the writing what the terms are supposed to represent, but making things more cohesive would improve the paper.

Additionally, in regards to notation it might be helpful to disambiguate the hard vs soft commitments with different variables, since it seems that \hat{x} is considered as both a soft and hard committed solution which gets “overwritten”. Mainly, this just changes that the problem in 152 refers to \hat{x} as a soft solution in the objective but hard solutions in the constraints.

[1] Wang, Kai, et al. "Automatically learning compact quality-aware surrogates for optimization problems." Advances in Neural Information Processing Systems 33 (2020): 9586-9596.

[2] Ferber, Aaron, et al. "Mipaal: Mixed integer program as a layer." Proceedings of the AAAI Conference on Artificial Intelligence. Vol. 34. No. 02. 2020.

[3] Shah, Sanket, et al. "Decision-focused learning without decision-making: Learning locally optimized decision losses." Advances in Neural Information Processing Systems 35 (2022): 1320-1332.

[4] Zharmagambetov, Arman, et al. "Landscape surrogate: Learning decision losses for mathematical optimization under partial information." Advances in Neural Information Processing Systems 36 (2024).

**Questions:**

1. Do the networks at stage t take in as input the revealed parameters from stage t-1 in addition to the features? Are those already considered as features? Are the standard baselines in BAS also given the previous step information as input?

**Limitations:**

The limitations are adequately addressed

---

> ### Author Rebuttal · Authors · 2024-08-07
>
> Thank you for your detailed review of our work. We respond to your concerns below.
>
> **Datasets**: Please see overall response. Furthermore, thank you for your suggestion of stock data. However, the datasets from the cited works are not suitable for our purposes either. This is because the cited works only deal with unknowns in the objective, meaning that across each data set, the constraints are always the same, so there is nothing to predict in our setting. As we mentioned in our overall response, the field is still in its infancy. Application domain practitioners have yet to collect publicly available benchmarks and datasets in the multi-stage P+O setting (which is a supervised learning problem), given that there were *no other works* handling our proposed learning setting. We thus hope that our work, by laying the foundations for this new framework, will serve as a "call to arms" for practitioners and researchers to collect datasets for further work on training methods.
> We will explicitly state this in the revised paper.
>
> **Incrementality compared with two-stage work**: We respectfully disagree with this assessment. Any multi-stage training approach, when restricted to $t=2$, will degenerate to a two-stage training method. In this sense, any multi-stage method will necessarily look like an unrolling of a two-stage method. Given that the problem setting itself is useful and enables a much wider class of applications, and the fact that we propose 3 *concrete* training methods, we believe that this line of work is valuable and *actionable* to disseminate to the community.
>
> **Substantial empirical improvement**: In Appendices F.1 and F.3 (referenced in lines 294 and 337), we gave per-simulation comparisons between our methods against "best among all standard regression methods" (BAS).
> Win rates can be directly computed from these plots by counting the number of simulations with ratios > 1.
> Here we present the win rate tables, showing that our methods outperform BAS in most simulations.
>
> Win rates for the production and sales problem.
> |Price group|Stage num|Baseline beats BAS|SCD beats BAS|PCD beats BAS|BAS is best|Baseline is best|SCD is best|PCD is best|
> |:-------------:|:---------:|:--------------------:|:-------------:|:-------------:|:-----------:|:----------------:|:-----------:|:-----------:|
> |Low-profit|4|93.33%|96.67%|86.67%|0.00%|3.33%|50.00%|46.67%|
> ||12|73.33%|100.00%|90.00%|0.00%|0.00%|66.67%|33.33%|
> |High-profit|4|66.67%|96.67%|73.33%|0.00%|23.33%|50.00%|26.67%|
> ||12|76.67%|100.00%|80.00%|0.00%|0.00%|63.33%|36.67%|
>
> Win rates for the 0-1 knapsack problem.
> |Capital|Stage_num|Trans_fee|Baseline beats BAS|SCD beats BAS|PCD beats BAS|BAS is best|Baseline is best|SCD is best|PCD is best|
> |:-------:|:---------:|:---------:|:--------------------:|:-------------:|:-------------:|:-----------:|:----------------:|:-----------:|:-----------:|
> |25|4|0.01|53.33%|86.67%|73.33%|3.33%|30.00%|46.67%|20.00%|
> |||0.1|70.00%|93.33%|90.00%|0.00%|33.33%|46.67%|20.00%|
> ||12|0.01|66.67%|93.33%|83.33%|0.00%|3.33%|73.33%|23.33%|
> |||0.1|83.33%|100.00%|96.67%|0.00%|0.00%|86.67%|13.33%|
> |50|4|0.01|60.00%|80.00%|66.67%|3.33%|23.33%|43.33%|30.00%|
> |||0.1|70.00%|96.67%|90.00%|0.00%|26.67%|43.33%|30.00%|
> ||12|0.01|70.00%|83.33%|83.33%|3.33%|0.00%|80.00%|16.67%|
> |||0.1|76.67%|100.00%|90.00%|0.00%|0.00%|93.33%|6.67%|
>
> Win rates for the nurse rostering problem.
> |Extra nurse payment|Baseline beats BAS|SCD beats BAS|PCD beats BAS|BAS is best |Baseline is best|SCD is best|PCD is best|PCD is best|
> |-----------------------|:--------------------:|:-------------:|:-------------:|:-----------:|:----------------:|:-----------:|:-----------:|:-----------:|
> |15|70.00%|70.00%|70.00%|10.00%|26.67%|40.00%|23.33%|46.67%|
> |20|73.33%|86.67%|80.00%|6.67%|10.00%|50.00%|33.33%|33.33%|
> |25|73.33%|96.67%|83.33%|3.33%|16.67%|43.33%|36.67%|26.67%|
> |30|73.33%|86.67%|76.67%|3.33%|6.67%|60.00%|30.00%|36.67%|
>
> **Necessity of coordinate-descent approach**: Yes, it is possible to train all networks simultaneously, e.g. by using ground truth parameters in place of prior and future stage predictions.
> However, intuitively, this is a worse approach than our methods, given the interdependency of the predictors: the performance of a predictor depends on the predictions and choices made in past and future stages.
> We did nonetheless explore this alternative training method before, with preliminary results, but the solution quality achieved was worse than the SCD and PCD methods.
>
> Mean post-hoc regrets and standard deviations for different training methods in the production and sales problem.
> |Price group|Low-profit||High-profit||
> |:--------------------:|:-------------:|:-------------:|:-------------:|:-------------:|
> |Stage_num|4|12|4|12|
> |SCD|293.78±99.21|488.72±127.62|505.24±89.55|887.38±250.55|
> |PCD|297.34±107.44|495.21±122.42|520.76±92.20|905.61±255.99|
> |train_simultaneously|300.28±103.85|509.07±129.93|523.38±88.52|925.35±223.17|
> |Baseline|305.26±100.88|515.80±137.67|526.77±104.99|935.03±263.47|
>
> The coordinate descent strategy in the paper was needed to capture the complex interactions between the networks in different stages, and crucial for the strong performance. We will include the results and discussion in the revised paper.
>
> **Off-by-one indices**: we double-checked, the indexing in the paper is correct, although we appreciate that the writing could be further clarified. Thank you for pointing this out and we will clean it up more in the paper.
>
> Line 142 refers to what happened just prior to stage $t$, and line 144 refers to what is happening during stage $t$ (so $\theta_t$ has been revealed at that point).
>
> Line 152: the soft solutions start at stage 0 in the sense that stage 0 generates a soft solution, but within the solution vector, the decision variables are between stages 1 to T (in the context of line 152 we only need to constrain the decisions in stages 1..t-1 for non-anticipativity).
>
> **Question**: Please see overall response.

---

### Official Review · Reviewer_GGkj · 2024-07-13

**Soundness:** 3
**Presentation:** 3
**Contribution:** 3
**Rating:** 5
**Confidence:** 1

**Summary:**

The paper proposes a new framework called Multi-Stage Predict+Optimize for tackling optimization problems with parameters revealed in multiple stages, rather than simultaneously. The authors develop three training algorithms for neural networks within this framework, particularly for mixed integer linear programs (MILPs). These algorithms include a baseline extension of prior work and two novel algorithms leveraging sequential and parallel coordinate descent methods. The paper demonstrates the efficacy of these methods through experiments on three benchmarks, showing superior learning performance over classical approaches.

**Strengths:**

1. The paper introduces the Multi-Stage Predict+Optimize framework, which extends traditional two-stage optimization methods to handle parameters revealed in multiple stages. This approach addresses a more realistic scenario in many real-world problems where information becomes available progressively.

2. The paper provides a detailed and rigorous development of the proposed methods. The theoretical foundations are well-explained, and the algorithms are clearly described, ensuring that the approach is both sound and replicable.

3. The experiments conducted on three benchmark problems demonstrate the effectiveness of the proposed methods. The results are presented in a clear and structured manner, highlighting the strengths of the Multi-Stage Predict+Optimize approach compared to classical techniques.

**Weaknesses:**

1. The proposed Multi-Stage Predict+Optimize framework, while an extension of existing two-stage methods, may not significantly differentiate itself from prior frameworks in practical applications. The core idea of updating parameter predictions and decisions in multiple stages is not sufficiently innovative compared to existing work in multi-stage stochastic optimization. To improve, the authors could more clearly highlight unique aspects and potential new applications of their framework that are not covered by existing methods.

2. Some sections of the paper, particularly those explaining the algorithms and theoretical foundations, may be dense and challenging for readers not deeply familiar with the subject. This can hinder the accessibility and broader understanding of the proposed methods. Simplifying the explanations and including more intuitive examples or visual aids can make the paper more accessible to a wider audience.

3. The benchmarks used for experimentation, while useful, may not fully represent the diversity of real-world applications. Additionally, the performance metrics and evaluation scenarios could be more comprehensive to cover various practical constraints and conditions. Expanding the experimental evaluation to include a wider variety of benchmarks and more complex real-world scenarios would strengthen the paper's claims.

**Questions:**

Please see the weakness. As I am unfamiliar with this topic, I am unsure of my judgment and may need to discuss it with other reviewers.

---

> ### Author Rebuttal · Authors · 2024-08-07
>
> Thank you for your review and your appreciation of the rigor in our work. Here, we address your concerns in the "Weaknesses" part of the review.
>
> **Comparison with existing Two-Stage framework**: We want to emphasize that the multi-stage framework *does* cover a much wider range of applications. Take for example the nurse rostering problem from one of our benchmarks: we adapted the scenario from Hu et al.'s Two-Stage paper. We in fact argue that Hu et al.'s modelling into the Two-Stage framework can be somewhat unrealistic: they essentially assume that shift schedules are necessarily set a whole week at a time, with no changes possible during the week, and with appointments for the entire week closing, say, weekly on the Sunday night prior. This is rigid and does not offer the (common) flexibility of daily appointment scheduling. By contrast, our new multi-stage framework captures such flexibility --- a weekly schedule is released to nurses at the beginning of the week, but last minute (or medium-term) changes can be made at a cost/penalty, reflecting the possibility of such practice in real businesses.
>
> **Comparison to Multi-Stage Stochastic Optimization**: Our work is very different from Multi-Stage Stochastic Optimization, with significantly disjoint challenges, as we detailed in Appendix A.3. Here, we highlight some of the main differences again.
>
> The most important distinction is that MSSO does not make predictions based on features, but instead assumes knowledge of the distribution of the true parameters. On the other hand, the multi-stage Predict+Optimize framework is a supervised learning problem with predictions made from features.
>
> The main challenge in MSSO, assuming that the distribution of true parameters we got is accurate, is primarily computational in the sense of the optimization problem -- how we can efficiently solve the complex, multi-stage optimization problem. The focus is on developing efficient algorithms and solution techniques to tackle the computational complexity of optimization.
>
> By contrast, the key challenge in the multi-stage Predict+Optimize framework is in the learning aspect -- how we should train prediction models for estimated parameters (which has its own computational challenges), so that when applied to the optimization problem, we can obtain good solutions.
> Once we have a prediction model, we just solve a sequence of (relatively simple, non-multi-stage) optimization problems, without the computational challenge of solving a multi-stage stochastic programming problem.
>
> In summary, the two frameworks are in fact very different technically, even though they look similar on the surface.
>
> **Benchmarks**: Please see the overall response.

---

> > ### Comment · Reviewer_GGkj · 2024-08-12
> >
> > Thanks for clarifying; I have raised my score to 5 but with less confidence since I am unfamiliar with this field.

---

> > > ### Author Response · Authors · 2024-08-12
> > >
> > > Thank you for taking the time to read and respond to our rebuttal. We are glad that our clarifications do address your concerns.

---

### Author Rebuttal · Authors · 2024-08-07

We thank the reviewers for the detailed, in-depth and constructive reviews. We are glad that reviewers recognize and appreciate that our work tackles an important problem, gives robust empirical evaluations, and that our paper is well-written.

In this overall response, we address some of the common reviewer concerns, and we also respond to individual comments in review-specific rebuttals. We hope to continue to engage with the reviewers in the author-reviewer discussion period -- we strongly believe in the value, importance and message of this work, and hope to convince you of the same.

**Benchmarks and datasets**: The benchmarks we used are more complex and sophisticated than the ones used by Hu et al.'s Two-Stage paper, not only due to the multi-stage nature, but also in other aspects of modelling (see our response to Reviewer Bz3V). In terms of datasets, we follow standard practice in the area that if we can't find directly relevant datasets, we use real datasets from an unrelated application domain in place. While one might reasonably question the choice somewhat, we point out that the Predict+Optimize field is very much still in its infancy, and our work is the *first* to propose a multi-stage framework in this supervised learning setting, also handling unknown parameters in constraints, which has been rarely studied in prior works. As such, for many application domains, there are simply no publicly available datasets suitable for evaluation in our problem setting (see our response to Reviewer aupq on why the cited stock market data is also unsuitable). We hope that our work, by contributing to the foundations of P+O, can serve as a "call to arms" along with prior works in the area, and encourage practitioners to start collecting/publishing data for other researchers to use for methodological research.

**Revealed parameters as a feature for later stage predictions**: In our current implementation, the network at stage $t$ in our proposed methods does not take the revealed parameters from stage $t-1$ as additional features as input, even though in principle we could write the framework to allow that. We did not include these revealed parameters because, in our preliminary experiments, including them does not really improve prediction quality while just increasing training time.

Here are the results from our preliminary experiments on the production and sales problem by incorporating the revealed parameters from stage $t-1$ as inputs to the stage $t$ networks:

Mean post-hoc regrets and standard deviations of training SCD with revealed parameters, training PCD with revealed parameters, SCD, PCD, and Baseline for the production and sales problem.
|     Price   group    |   Low-profit  |               |  High-profit  |               |
|:--------------------:|:-------------:|:-------------:|:-------------:|:-------------:|
|       Stage_num      |       4       |       12      |       4       |       12      |
| SCD_with_revealed_param    |  294.44±81.72 | 488.93±116.51 |  507.30±63.74 | 890.48±153.28 |
| PCD_with_revealed_param    |  299.73±82.91 | 498.32±123.25 |  521.82±74.86 | 911.04±185.57 |
|          SCD         |  293.78±99.21 | 488.72±127.62 |  505.24±89.55 | 887.38±250.55 |
|          PCD         | 297.34±107.44 | 495.21±122.42 |  520.76±92.20 | 905.61±255.99 |
|       Baseline       | 305.26±100.88 | 515.80±137.67 | 526.77±104.99 | 935.03±263.47 |

The table shows that the performance using this expanded input did not improve over the results of our proposed approaches. We will include these preliminary comparison results and discussion in the paper if accepted.

---

### Decision · Program_Chairs · 2024-09-25

**Decision:**

Accept (poster)

**Comment:**

This paper considers the problem of learning unknown parameters for multi-stage optimization problems which are gradually known at each stage. The latent parameters are predicted to compute soft decisions; the stage's unknown parameters become known; practitioner can modify the soft decision to a hard decision for some penalty; and once the hard decision is fixed, the next stage begin. The post hoc regret is computed once all the stages are done. The overall goal is to train models to predict the latent parameters by minimizing the overall regret. The paper generalizes previous end-to-end training methods for two-stage problems.

This is a borderline paper based on the reviews and discussion. One of the reviewer who gave a initial negative review could not participate in the discussion due to emergency situation (the author response seems to mostly address their critical comments). Here are strengths and weaknesses based on my reading and assessment.

(++) Well-motivated new problem setting

(+) The proposed training approaches are reasonable but not particularly innovative

(-) The evaluation is on synthetic problems. It would have been nice to have at least one real-world problem to demonstrate the practical utility

I'm recommending accepting the paper with the hope that this new problem setting may inspire new research by both AI researchers and practitioners to expand the knowledge of decision-focused learning paradigm.